# Robust $\phi$-Divergence MDPs

**Chin Pang Ho**
City University of Hong Kong
`clint.ho@cityu.edu.hk`

**Marek Petrik**
University of New Hampshire
`mpetrik@cs.unh.edu`

**Wolfram Wiesemann**
Imperial College London
`ww@imperial.ac.uk`

## Abstract

In recent years, robust Markov decision processes (MDPs) have emerged as a prominent modeling framework for dynamic decision problems affected by uncertainty. In contrast to classical MDPs, which only account for *stochasticity* by modeling the dynamics through a stochastic process with a known transition kernel, robust MDPs additionally account for *ambiguity* by optimizing in view of the most adverse transition kernel from a prescribed ambiguity set. In this paper, we develop a novel solution framework for robust MDPs with $s$-rectangular ambiguity sets that decomposes the problem into a sequence of robust Bellman updates and simplex projections. Exploiting the rich structure present in the simplex projections corresponding to $\phi$-divergence ambiguity sets, we show that the associated $s$-rectangular robust MDPs can be solved substantially faster than with state-of-the-art commercial solvers as well as a recent first-order solution scheme, thus rendering them attractive alternatives to classical MDPs in practical applications.

## 1 Introduction

Markov decision processes (MDPs) are a flexible and popular framework for dynamic decision-making problems and reinforcement learning [39, 48]. A practical limitation of the standard MDP model is that it assumes the model parameters, such as transition probabilities and rewards, to be known exactly. In reinforcement learning and other applications, these parameters must be estimated from sampled data, which introduces estimation errors. Optimal MDP solutions, referred to as policies, are well known to be sensitive to errors and may fail catastrophically when deployed [25, 56].

Robust MDPs (RMDPs) mitigate the sensitivity of MDPs to estimation errors by computing a policy that is optimal for the worst plausible realization of the transition probabilities. This set of plausible transition probabilities is known as the *ambiguity set*. Most prior work considers ambiguity sets that are rectangular. In this work, we focus on *s-rectangular ambiguity sets*, which assume that the worst transition probabilities are chosen independently in each state [25, 56]. While several other models of rectangularity have been studied [9, 13, 21, 28], $s$-rectangular ambiguity sets are popular due to their generality and their performance both in-sample and out-of-sample [56]. However, even though polynomial-time algorithms based on dynamic programming concepts have been developed for $s$-rectangular sets, those algorithms may be too slow in practice. Solving RMDPs requires the solution of a convex optimization problem in every step of value or policy iteration, which can become prohibitively slow even in moderatly sized problems with 100s of states [5, 9, 14, 19].

Motivated by the difficulty of solving RMDPs, several fast algorithms have been proposed for $s$-rectangular RMDPs [5, 9, 14, 19]. The preponderance of the earlier work has focused on ambiguity sets defined in terms of $L_1$- and $L_\infty$-norms. These ambiguity sets are polyhedral, and they can be analyzed using linear programming techniques which offer fruitful avenues to exploit the structure inherent to those sets. However, recent statistical studies point to the superior solution quality offered by nonlinear ambiguity sets defined in terms of the Kullback-Leibler (KL) divergence, the $L_2$-norm and other metrics [17]. Linear optimization solvers are not applicable to RMDPs with $s$-rectangular ambiguity sets defined in terms of non-polyhedral ambiguity sets, as the corresponding optimization

36th Conference on Neural Information Processing Systems (NeurIPS 2022).

problems are in general convex conic programs (e.g., exponential cone program in the case of KL divergence); thus, they are currently solved using first-order methods [14] or general convex conic solvers such as MOSEK [3], which tend to be complex, closed-source and slow.

As our main contribution, we propose a new suite of fast algorithms for solving RMDPs with $\phi$-divergence constrained $s$-rectangular ambiguity sets. $\phi$-divergences, also known as f-divergences, constitute a generalization of the KL divergence that encompasses the Burg entropy as well as the $L_1$- and weighted $L_2$-norms as special cases [4, 6]. Moreover, $\phi$-divergence ambiguity sets benefit from rigorous statistical performance guarantees, and they are optimal among all (known and unknown) data-driven optimization paradigms for certain types of worst-case out-of-sample performance guarantees [35]. The radii of $\phi$-divergence ambiguity sets can be selected either via cross-validation or via statistical bounds [26, 32, 55]. Robust MDPs with $\phi$-divergence sets are challenging and unexplored for both $(s, a)$-rectangular and $s$-rectangular ambiguity sets. Solving $\phi$-divergence RMDPs using value iteration requires the solution of seemingly unstructured min-max problems. Our main insight is that these min-max problems can be reduced to a small number of highly structured projection problems onto a probability simplex. We use this insight to develop tailored solution schemes for the projection problems corresponding to several popular $\phi$-divergence ambiguity sets, which in turn give rise to efficient solution methods for the respective RMDPs. Ignoring tolerances, our algorithms achieve an overall $\mathcal{O}(S^2 \cdot A \log A)$ or $\mathcal{O}(S^2 \log S \cdot A)$ time complexity to compute the robust Bellman operator, where $S$ and $A$ denote the numbers of states and actions, respectively. Since the evaluation of a non-robust Bellman operator requires a runtime of $\mathcal{O}(S^2 \cdot \log A)$, our algorithms only incur an additional logarithmic overhead to account for robustness in the transition probabilities. This computational complexity compares favorably with the larger time complexity of a recent first-order solution scheme for KL divergence-constrained $s$-rectangular RMDPs (which we will elaborate on later in the paper) as well as a minimum complexity of $\mathcal{O}(S^{4.5} \cdot A)$ for the naïve solution with state-of-the-art interior-point algorithms. Our framework is general enough to readily accommodate for $\phi$-divergences that have not been studied previously in the context of $s$-rectangular ambiguity sets, such as the Burg entropy and the $\chi^2$-distance. For other $\phi$-divergences, such as the $L_1$-norm, our framework results in the same complexity at substantially simplified proofs.

The algorithms developed in this paper speed up the computation of robust Bellman updates, and so they can be used in combination with a variety of RMDP solution schemes. In particular, they can be used to accelerate the standard robust value iteration, policy iteration, modified policy iteration [22] and partial policy iteration [19]. They can also be combined with a first order gradient method [14] that has been introduced recently. In addition, fast algorithms for computing the Bellman operator also play a crucial role when scaling robust algorithms to value function approximation [50], model-free reinforcement learning [33, 42], and robust policy gradients [49]. In this paper, we focus on the model-based setting, which is currently under active study [24, 29, 31] and has many important real-life applications [12, 20, 58]; moreover, it also serves as an important building block to constructing model-free algorithms. While this paper focuses on the $s$-rectangular ambiguity sets, the proposed algorithms in this paper can also be applied to the special case of $(s, a)$-rectangular ambiguity sets.

The remainder of the paper proceeds as follows. Section 2 reviews relevant prior work and Section 3 describes our basic RMDP setting. Then, Section 4 shows how the robust Bellman operator for a large class of ambiguity sets can be reduced to a sequence of structured projections onto a simplex. We describe novel algorithms for efficiently computing the simplex projections for several $\phi$-divergences in Section 5. Finally, Section 6 presents experimental results that compare the runtime of our algorithms with general conic solvers as well as a recent first-order optimization algorithm [14].

**Notation.** We denote by $\mathbf{e}$ the vector of all ones, whose context determines its dimension. We refer to the probability simplex in $\mathbb{R}^n$ by $\Delta_n = \{\boldsymbol{p} \in \mathbb{R}_+^n : \mathbf{e}^\top \boldsymbol{p} = 1\}$. For $\boldsymbol{x} \in \mathbb{R}^n$, we let $\min\{\boldsymbol{x}\} = \min\{x_i : i = 1, \ldots, n\}$ (similar for the maximum operator), and we define $[\boldsymbol{x}]_+ \in \mathbb{R}_+^n$ component-wise as $([\boldsymbol{x}]_+)_i = \max\{x_i, 0\}$, $i = 1, \ldots, n$. We refer to the conjugate of a function $f : \mathbb{R}^n \to \mathbb{R}$ by $f^\star(\boldsymbol{y}) = \sup\{\boldsymbol{y}^\top \boldsymbol{x} - f(\boldsymbol{x}) : \boldsymbol{x} \in \mathbb{R}^n\}$. Random variables are indicated by a tilde.

## 2 Related Work

While RMDPs have been studied since the seventies [45], they have witnessed significant recent interest due to their widespread adoption in applications ranging from assortment optimization [43], medical decision-making [12, 62] and hospital operations management [16], production planning [58]

and energy systems [20] to model predictive control [10], aircraft collision avoidance [23], wireless communications [57] and the robustification against approximation errors in aggregated MDPs [37].

Efficient implementations of the robust value iteration have been first proposed by [11, 21, 32] for RMDPs with $(s, a)$-rectangular ambiguity sets, where the worst transition probabilities are considered separately for each state and action. The authors study ambiguity sets that bound the distance of the transition probabilities to some nominal distribution in terms of finite scenarios, interval matrix bounds, ellipsoids, the relative entropy, the KL divergence and maximum a posteriori models. Subsequently, similar methods have been developed by [57] for interval matrix bounds as well as likelihood uncertainty models, by [37] for 1-norm ambiguity sets as well as by [62] for interval matrix bounds intersected with a budget constraint. All of these contributions have in common that they focus on $(s, a)$-rectangular ambiguity sets where the existence of optimal deterministic policies is guaranteed, and it is not clear how they could be extended to the more general class of $s$-rectangular ambiguity sets where all optimal policies may be randomized.

In contrast to $(s, a)$-rectangular ambiguity sets, $s$-rectangular ambiguity sets restrict the conservatism among transition probabilities corresponding to different actions in the same state, which tends to lead to a superior performance in data-driven settings. [56] solve the subproblems arising in the robust value iteration of an $s$-rectangular RMDP as linear or conic optimization problems using commercial off-the-shelf solvers. Despite their polynomial-time complexity, general-purpose solvers cannot exploit the structure present in these subproblems, which renders them suitable primarily for small problem instances. More efficient tailored solution methods for $s$-rectangular RMDPs have subsequently been developed by [5, 18, 19]. [18] develop a homotopy continuation method for RMDPs with $(s, a)$-rectangular and $s$-rectangular weighted 1-norm ambiguity sets, while [5] adapt the algorithm of [18] to unweighted $\infty$-norm ambiguity sets. [19] embed the algorithms of [18] in a partial policy iteration, which generalizes the robust modified policy iteration proposed by [22] for $(s, a)$-rectangular RMDPs to $s$-rectangular RMDPs.

While the present paper focuses on the robust value iteration for ease of exposition, we note that our algorithms can also be combined with the partial policy iteration of [19] to obtain further speedups. [9] establish a relationship between $s$-rectangular RMDPs and twice regularized MDPs, which they subsequently use to propose efficient Bellman updates for a modified policy iteration. While their approach can solve RMDPs in almost the same time as a classical non-robust MDPs, the obtained policies can be conservative as the worst-case transition probabilities are not restricted to reside in a probability simplex and, therefore, may be negative and/or add up to more or less than 1. Finally, [14] propose a first-order framework for RMDPs with $s$-rectangular KL and spherical ambiguity sets that interleaves primal-dual first-order updates with approximate value iteration steps. The authors show that their algorithms outperform a robust value iteration that solves the emerging subproblems using state-of-the-art commercial solvers. We compare our solution method for KL ambiguity sets with the approach proposed by [14] in terms of its theoretical complexity and numerical runtimes.

While this paper exclusively studies $s$-rectangular uncertainty sets, alternative generalizations of $(s, a)$-rectangular ambiguity sets have been proposed in the literature as well. For example, [28] consider $k$-rectangular ambiguity sets where the transition probabilities of different states can be coupled, [13] study factor ambiguity model ambiguity sets where the transition probabilities depend on a small number of underlying factors, and [51] construct ambiguity sets that bound marginal moments of state-action features defined over entire MDP trajectories. Moving beyond the model-based setting, there is also an active line of research on robust reinforcement learning, such as least squares policy iteration [33], analysis on sample complexity [34], robust Q-learning and robust TDC algorithms [42, 53], and robust policy gradient [54]. We also note the papers [7, 15, 60] which study the related problem of *distributionally* robust MDPs whose transition probabilities are themselves regarded as random objects that are drawn from distributions which are only partially known. The connections between RMDPs and multi-stage stochastic programs as well as distributionally robust problems are explored further by [44, 46, 47].

## 3 Preliminaries

**Robust MDPs**  We study RMDPs with a finite state space $\mathcal{S} = \{1, \ldots, S\}$ and a finite action space $\mathcal{A} = \{1, \ldots, A\}$. We assume an infinite planning horizon, but all of our results immediately extend to a finite time horizon. Without loss of generality, we assume that every action $a \in \mathcal{A}$ is admissible

in every state $s \in \mathcal{S}$. The RMDP starts in a random initial state $\tilde{s}_0$ that follows the known probability distribution $\boldsymbol{p}^0$ from the probability simplex $\Delta_S$ in $\mathbb{R}^S$. If action $a \in \mathcal{A}$ is taken in state $s \in \mathcal{S}$, then the RMDP transitions randomly to the next state according to the conditional probability distribution $\boldsymbol{p}_{sa} \in \Delta_S$. We condense the transition probabilities $\boldsymbol{p}_{sa}$ to the tensor $\boldsymbol{p} \in (\Delta_S)^{S \times A}$. The transition probabilities are only known to reside in a non-empty, compact ambiguity set $\mathcal{P} \subseteq (\Delta_S)^{S \times A}$. For a transition from state $s \in \mathcal{S}$ to state $s' \in \mathcal{S}$ under action $a \in \mathcal{A}$, the decision maker receives an expected reward of $r_{sas'} \in \mathbb{R}_+$. As with the transition probabilities, we condense these rewards to the tensor $\boldsymbol{r} \in \mathbb{R}_+^{S \times A \times S}$. Without loss of generality, we assume that all rewards are non-negative.

We denote by $\Pi = (\Delta_A)^S$ the set of all stationary (i.e., time-independent) randomized policies. A policy $\boldsymbol{\pi} \in \Pi$ takes action $a \in \mathcal{A}$ in state $s \in \mathcal{S}$ with probability $\pi_{sa}$. The transition probabilities $\boldsymbol{p} \in \mathcal{P}$ and the policy $\boldsymbol{\pi} \in \Pi$ induce a stochastic process $\{(\tilde{s}_t, \tilde{a}_t)\}_{t=0}^{\infty}$ on the space $(\mathcal{S} \times \mathcal{A})^{\infty}$ of sample paths. We refer by $\mathbb{E}^{\boldsymbol{p},\boldsymbol{\pi}}$ to expectations with respect to this process. The decision maker is risk-neutral but ambiguity-averse and wishes to maximize the worst-case expected total reward under a discount factor $\lambda \in (0,1)$,

$$\max_{\boldsymbol{\pi} \in \Pi} \min_{\boldsymbol{p} \in \mathcal{P}} \mathbb{E}^{\boldsymbol{p},\boldsymbol{\pi}} \left[ \sum_{t=0}^{\infty} \lambda^t \cdot r_{\tilde{s}_t, \tilde{a}_t, \tilde{s}_{t+1}} \mid \tilde{s}_0 \sim \boldsymbol{p}^0 \right]. \tag{1}$$

Note that the maximum and minimum in (1) are both attained by the Weierstrass theorem since $\Pi$ and $\mathcal{P}$ are non-empty and compact, while the objective function is finite since $\lambda < 1$.

**Rectangular Ambiguity Sets** For general ambiguity sets $\mathcal{P}$, evaluating the inner minimization in (1) is NP-hard even if the policy $\boldsymbol{\pi} \in \Pi$ is fixed [56]. For these reasons, much of the research on RMDPs and their applications has focused on rectangular ambiguity sets. Among the most general rectangular ambiguity sets are the $s$-rectangular ambiguity sets $\mathcal{P}$ satisfying

$$\mathcal{P} = \left\{ \boldsymbol{p} \in (\Delta_S)^{S \times A} : \boldsymbol{p}_s \in \mathcal{P}_s \; \forall s \in \mathcal{S} \right\}, \quad \text{where} \quad \mathcal{P}_s \subseteq (\Delta_S)^A, \; s \in \mathcal{S},$$

see [25, 56, 59, 61]. In contrast to the simpler class of $(s,a)$-rectangular ambiguity sets, $s$-rectangular ambiguity sets restrict the choice of transition probabilities $\boldsymbol{p}_{s1}, \ldots, \boldsymbol{p}_{sA}$ corresponding to different actions $a$ applied in the same state $s$. This limits the conservatism of the resulting RMDP (1) and typically leads to a better performance of the optimal policy [56]. Although Bellman's optimality principle extends to $s$-rectangular RMDPs and there is always an optimal stationary policy, all optimal policies of an $s$-rectangular RMDP may be randomized.

We study a new general class of $s$-rectangular ambiguity sets that can be expressed as

$$\mathcal{P}_s = \left\{ \boldsymbol{p}_s \in (\Delta_S)^A : \sum_{a \in \mathcal{A}} d_a(\boldsymbol{p}_{sa}, \overline{\boldsymbol{p}}_{sa}) \leq \kappa \right\}, \tag{2}$$

where $\kappa \in \mathbb{R}_+$ is the *uncertainty budget* and the distance functions $d_a(\boldsymbol{p}_{sa}, \overline{\boldsymbol{p}}_{sa})$, $a \in \mathcal{A}$, are *$\phi$-divergences* (also known as *f-divergences*) satisfying

$$d_a(\boldsymbol{p}_{sa}, \overline{\boldsymbol{p}}_{sa}) = \sum_{s' \in \mathcal{S}} \overline{p}_{sas'} \phi \left( \frac{p_{sas'}}{\overline{p}_{sas'}} \right).$$

Here, $\phi \colon \mathbb{R}_+ \to \mathbb{R}_+$ is a convex function satisfying $\phi(1) = 0$. Intuitively, a $\phi$-divergence measures the distance between two probability distributions. With an appropriate choice of $\phi$, it generalizes other metrics including the KL divergence, the Burg entropy, $L_1$- and $L_2$-norms and others [4, 6]. Table 1 reports some popular $\phi$-divergences that we study in this paper. The variation distance coincides with the $L_1$-based $s$-rectangular ambiguity sets studied in earlier work [18, 19]. Note that although we assume that $\phi$ is the same for different state-action pairs, the proposed approach also works for the more general case where $d_a(\boldsymbol{p}_{sa}, \overline{\boldsymbol{p}}_{sa}) = \sum_{s' \in \mathcal{S}} \overline{p}_{sas'} \phi_{sas'}(p_{sas'}/\overline{p}_{sas'})$.

**Robust Value Iteration** A standard approach for computing the optimal value and the optimal policy of an RMDP (1) is the robust value iteration [21, 32, 25, 56]: Starting with an initial estimate $\boldsymbol{v}^0 \in \mathbb{R}^S$ of the state-wise optimal value to-go, we conduct robust Bellman iterations of the form $\boldsymbol{v}^{t+1} \leftarrow \mathfrak{J}(\boldsymbol{v}^t)$, $t = 0, 1, \ldots$, where the robust Bellman operator $\mathfrak{J}$ is defined component-wise as

$$[\mathfrak{J}(\boldsymbol{v})]_s = \max_{\boldsymbol{\pi}_s \in \Delta_A} \min_{\boldsymbol{p}_s \in \mathcal{P}_s} \sum_{a \in \mathcal{A}} \pi_{sa} \cdot \boldsymbol{p}_{sa}^{\top} (\boldsymbol{r}_{sa} + \lambda \boldsymbol{v}) \qquad \forall s \in \mathcal{S}. \tag{3}$$

| Divergence | $d_a(\boldsymbol{p}_{sa}, \overline{\boldsymbol{p}}_{sa})$ | $\phi(t)$ | Complexity of $\mathfrak{J}$ | Prior Art |
|---|---|---|---|---|
| Kullback-Leibler | $\sum_{s'} p_{sas'} \log\left(\frac{p_{sas'}}{\overline{p}_{sas'}}\right)$ | $t \log t - t + 1$ | $\mathcal{O}(S^2 A \log A)$ | $\mathcal{O}(\ell^2 S^2 A)$ |
| Burg Entropy | $\sum_{s'} \overline{p}_{sas'} \log\left(\frac{\overline{p}_{sas'}}{p_{sas'}}\right)$ | $-\log t + t - 1$ | $\mathcal{O}(S^2 A \log A)$ | no poly-time |
| Variation Distance | $\sum_{s'} |p_{sas'} - \overline{p}_{sas'}|$ | $|t-1|$ | $\mathcal{O}(S^2 \log SA)$ | $\mathcal{O}(S^2 \log SA)$ |
| $\chi^2$-Distance | $\sum_{s'} \frac{(p_{sas'} - \overline{p}_{sas'})^2}{\overline{p}_{sas'}}$ | $(t-1)^2$ | $\mathcal{O}(S^2 \log SA)$ | $\mathcal{O}(S^{4.5} A)$ |

Table 1: Summary of the $\phi$-divergences studied in this paper, together with the complexity of our robust Bellman operator $\mathfrak{J}$ (applied across all states $s \in \mathcal{S}$) as well as the best known results from the literature. The complexity estimates omit constants and tolerances that are reported in Section 5 of the paper. '$\ell$', where present, refers to the number of Bellman iterations conducted so far.

This yields the optimal value $\boldsymbol{p}^{0\top} \boldsymbol{v}^\star$, where the limit $\boldsymbol{v}^\star = \lim_{t \to \infty} \boldsymbol{v}^t$ is approached component-wise at a geometric rate. The optimal policy $\boldsymbol{\pi}^\star \in \Pi$, finally, is recovered state-wise via

$$\boldsymbol{\pi}_s^\star \in \arg\max_{\boldsymbol{\pi}_s \in \Delta_A} \min_{\boldsymbol{p}_s \in \mathcal{P}_s} \sum_{a \in \mathcal{A}} \pi_{sa} \cdot \boldsymbol{p}_{sa}^\top (\boldsymbol{r}_{sa} + \lambda \boldsymbol{v}^\star) \qquad \forall s \in \mathcal{S}.$$

## 4  Robust Bellman Updates via Simplex Projections

In this section, we show that the robust Bellman operator $\mathfrak{J}$ reduces to a generalized projection problem. This reduction is important because it underlies our fast algorithms for computing $\mathfrak{J}$.

At the core of the robust value iteration is the solution of the max-min problem (3). By applying the minimax theorem, the right-hand side of (3) equals to

$$
\begin{aligned}
\text{minimize} \quad & \max_{a \in \mathcal{A}} \left\{ \boldsymbol{p}_{sa}^\top (\boldsymbol{r}_{sa} + \lambda \boldsymbol{v}) \right\} \\
\text{subject to} \quad & \sum_{a \in \mathcal{A}} d_a(\boldsymbol{p}_{sa}, \overline{\boldsymbol{p}}_{sa}) \leq \kappa \\
& \boldsymbol{p}_s \in (\Delta_S)^A.
\end{aligned}
\tag{4}
$$

The above optimization problem can be solved via bisection on its objective value; that is, we seek the lowest possible $\beta$ such that $\boldsymbol{p}_{sa}^\top (\boldsymbol{r}_{sa} + \lambda \boldsymbol{v}) \leq \beta$, for each $a \in \mathcal{A}$ where $\boldsymbol{p}_s$ satisfies the constraints in (4). For any given $\beta$ in the bisection method, we check whether such $\boldsymbol{p}_s$ exists by solving the following generalized $d_a$-projection problem of the nominal transition probabilities $\overline{\boldsymbol{p}}_{sa}$:

$$
\mathfrak{P}(\overline{\boldsymbol{p}}_{sa}; \boldsymbol{b}, \beta) = \begin{bmatrix} \text{minimize} & d_a(\boldsymbol{p}_{sa}, \overline{\boldsymbol{p}}_{sa}) \\ \text{subject to} & \boldsymbol{b}^\top \boldsymbol{p}_{sa} \leq \beta \\ & \boldsymbol{p}_{sa} \in \Delta_S \end{bmatrix}.
\tag{5}
$$

Here, $\boldsymbol{p}_{sa} \in \Delta_S$ are the decision variables and $\overline{\boldsymbol{p}}_{sa} \in \Delta_S$, $\boldsymbol{b} \in \mathbb{R}_+^S$ and $\beta \in \mathbb{R}_+$ are parameters. For any fixed $\beta$, we compute $\sum_{a \in \mathcal{A}} \mathfrak{P}(\overline{\boldsymbol{p}}_{sa}; \boldsymbol{r}_{sa} + \lambda \boldsymbol{v}, \beta)$ and construct $\boldsymbol{p}_s \in (\Delta_S)^A$ where $\boldsymbol{p}_{sa}$ is the optimal solution of the associated projection problem $\mathfrak{P}(\overline{\boldsymbol{p}}_{sa}; \boldsymbol{b}, \beta)$. We can then distinguish between the following two cases:

1. If $\sum_{a \in \mathcal{A}} \mathfrak{P}(\overline{\boldsymbol{p}}_{sa}; \boldsymbol{r}_{sa} + \lambda \boldsymbol{v}, \beta) \leq \kappa$, then the constructed $\boldsymbol{p}_s$ is a feasible solution to (5). Therefore, $\beta$ upper bounds the optimal objective value.

2. If $\sum_{a \in \mathcal{A}} \mathfrak{P}(\overline{\boldsymbol{p}}_{sa}; \boldsymbol{r}_{sa} + \lambda \boldsymbol{v}, \beta) > \kappa$, then there is no feasible $\boldsymbol{p}_s \in (\Delta_S)^A$ such that the objective value attains $\beta$ or less. Therefore, $\beta$ lower bounds the optimal objective value.

As a consequence of the cases above, one can compute the robust Bellman update (3) efficiently by bisection if the projection problem (5) can be solved efficiently. The proof of Theorem 1 describes further details needed to implement this algorithm, including the initial upper and lower bounds on $\beta$ and bounds on the precision needed when solving the projection problems.

Note that problem (5) is infeasible if and only if $\min\{\boldsymbol{b}\} > \beta$. Moreover, problem (5) is trivially solved by $\overline{\boldsymbol{p}}_{sa}$ with an optimal objective value of 0 whenever $\boldsymbol{b}^\top \overline{\boldsymbol{p}}_{sa} \leq \beta$. To avoid these trivial cases,

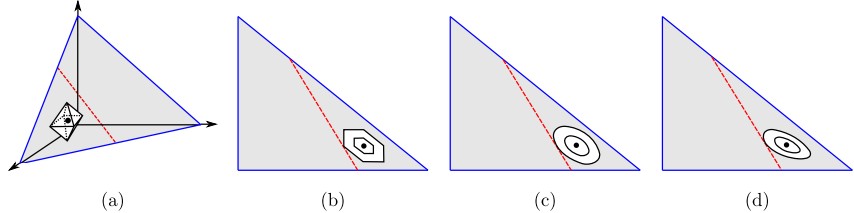

(a)       (b)       (c)       (d)

Figure 1: The generalized $d_a$-projection problem (5) in $S = 3$ dimensions (a) and two-dimensional projections for the variation distance (b), the $\chi^2$-distance (c) and the KL divergence (d). The gray shaded areas represent the probability simplex $\Delta_S$, the red dashed lines show the boundary of the intersection of the halfspace $\boldsymbol{b}^\top \boldsymbol{p}_{sa} \leq \beta$ with the probability simplex, and the white shapes illustrate contour lines centered at the nominal transition probabilities $\overline{\boldsymbol{p}}_{sa}$.

we assume throughout the paper that $\min\{\boldsymbol{b}\} \leq \beta$ and $\boldsymbol{b}^\top \overline{\boldsymbol{p}}_{sa} > \beta$. We illustrate the feasible region and optimal solution to problem (5) for different $\phi$-divergences in Figure 1.

Our generalized $d_a$-projection (5) relates to the rich literature on projections onto simplices, which we review in the next section. In fact, our algorithms in the next section solve a variant of the simplex projection problem that is restricted by an additional inequality constraint. We therefore believe that our algorithms may find additional applications outside the RMDP literature.

In the following, we say that for a given estimate $\boldsymbol{v}^t \in \mathbb{R}^S$ of the optimal value function, the robust Bellman iteration (3) is solved to $\epsilon$-accuracy by any $\boldsymbol{v}^{t+1} \in \mathbb{R}^S$ satisfying $\|\boldsymbol{v}^{t+1} - \mathfrak{J}(\boldsymbol{v}^t)\|_\infty \leq \epsilon$. We seek $\epsilon$-optimal solutions because our ambiguity sets are nonlinear and hence the exact Bellman iterate $\mathfrak{J}(\boldsymbol{v}^t)$ may be irrational even if $\boldsymbol{v}^t$ is rational. To simplify the exposition, we define $\overline{R} = [1 - \lambda]^{-1} \cdot \max\{r_{sas'} : s, s' \in \mathcal{S}, a \in \mathcal{A}\}$ as an upper bound on all $[\mathfrak{J}(\boldsymbol{v})]_s$, $\boldsymbol{v} \leq \boldsymbol{v}^\star$ and $s \in \mathcal{S}$.

For divergence-based ambiguity sets, the projection problem (5) is generically nonlinear and can hence not be expected to be solved to exact optimality. To account for this additional complication, we say that for a given $\overline{\boldsymbol{p}}_{sa} \in \Delta_S$, $\boldsymbol{b} \in \mathbb{R}_+^S$ and $\beta \in \mathbb{R}_+$, the generalized $d_a$-projection $\mathfrak{P}(\overline{\boldsymbol{p}}_{sa}; \boldsymbol{b}, \beta)$ is solved to $\delta$-accuracy by any pair $(\underline{d}, \overline{d}) \in \mathbb{R}^2$ satisfying $\mathfrak{P}(\overline{\boldsymbol{p}}_{sa}; \boldsymbol{b}, \beta) \in [\underline{d}, \overline{d}]$ and $\overline{d} - \underline{d} \leq \delta$.

**Theorem 1.** *Assume that the generalized $d_a$-projection (5) can be computed to any accuracy $\delta > 0$ in time $\mathcal{O}(h(\delta))$. Then the robust Bellman iteration (3) can be computed to any accuracy $\epsilon > 0$ in time $\mathcal{O}(AS \cdot h(\epsilon\kappa/[2A\overline{R} + A\epsilon]) \cdot \log[\overline{R}/\epsilon])$.*

Theorem 1 reduces the evaluation of the robust Bellman iterator $\mathfrak{J}$, which involves the solution of a max-min optimization problem over an $s$-rectangular ambiguity set that couples all actions $a \in \mathcal{A}$, to a sequence of much simpler and highly structured projection problems that are no longer coupled across different actions $a \in \mathcal{A}$. The next section describes efficient solution schemes for the projection problem (5) in the context of several $\phi$-divergence ambiguity sets. The runtimes of these solution schemes are summarized in Table 1. Note that the evaluation of a non-robust Bellman operator requires a runtime of $\mathcal{O}(S^2 \cdot \log A)$, which implies that our algorithms only incur an additional logarithmic overhead to account for robustness in the transition probabilities.

## 5    Fast Projections on $\phi$-Divergence Simplices

We next describe fast algorithms for computing generalized projections onto the probability simplex. Combined with the results from Section 4, these algorithms can be used to efficiently compute the robust Bellman operator. Note that some $\phi$-divergences, such as the KL divergence and the $\chi^2$-distance, imply that if $\overline{p}_{sas'} = 0$ for some $s, s' \in \mathcal{S}$ and $a \in \mathcal{A}$, then $p_{sas'} = 0$ for all $\boldsymbol{p}_{sa} \in \Delta_S$ with $d_a(\boldsymbol{p}_{sa}, \overline{\boldsymbol{p}}_{sa}) < \infty$, and thus we can remove indices $s'$ with $\overline{p}_{sas'} = 0$. For other $\phi$-divergences, such as the Burg entropy and the variation distance, one can readily verify that our results remain valid no matter whether $\overline{\boldsymbol{p}}_{sa} > \boldsymbol{0}$ or not, but the formulations and proofs require additional case distinctions and/or limit arguments. To simplify the exposition, we therefore assume that $\overline{\boldsymbol{p}}_{sa} > \boldsymbol{0}$.

**Proposition 1.** *For the distance function $d_a(\boldsymbol{p}_{sa}, \overline{\boldsymbol{p}}_{sa}) = \sum_{s' \in \mathcal{S}} \overline{p}_{sas'} \cdot \phi\left(\frac{p_{sas'}}{\overline{p}_{sas'}}\right)$, the optimal value of the projection problem* (5) *equals the optimal value of the bivariate convex problem*

$$
\begin{aligned}
\text{maximize} \quad & -\beta\alpha + \zeta - \sum_{s' \in \mathcal{S}} \overline{p}_{sas'} \phi^{\star}(-\alpha b_{s'} + \zeta) \\
\text{subject to} \quad & \alpha \in \mathbb{R}_+, \ \ \zeta \in \mathbb{R}.
\end{aligned}
\tag{6}
$$

Proposition 1 reduces the $S$-dimensional projection problem (5) to a two-dimensional optimization problem over the dual variables $\alpha$ and $\zeta$. In the following, we show that for the $\phi$-divergences from Table 1, problem (6) can be further simplified to univariate convex optimization problems that can be solved efficiently via bisection, binary search or sorting.

## 5.1 Kullback-Leibler Divergence

We first show that for the KL divergence $\phi(t) = t \log t - t + 1$, the reduced projection problem (6) can be further simplified to a univariate convex optimization problem.

**Proposition 2.** *For the KL divergence $\phi(t) = t \log t - t + 1$, the optimal value of the projection problem* (5) *equals the optimal value of the univariate convex problem*

$$
\underset{\alpha \in \mathbb{R}_+}{\text{maximize}} \quad -\beta\alpha - \log\left(\sum_{s' \in \mathcal{S}} \overline{p}_{sas'} \cdot \mathrm{e}^{-\alpha b_{s'}}\right).
\tag{7}
$$

We next show that the univariate optimization problem (2) admits an efficient solution via bisection.

**Theorem 2.** *If $\beta \geq \min\{\boldsymbol{b}\} + \omega$ for some $\omega > 0$, then the projection problem* (5) *can be solved to any $\delta$-accuracy in time $\mathcal{O}(S \cdot \log[\max\{\boldsymbol{b}\} \cdot \log(\min\{\overline{\boldsymbol{p}}\}^{-1})/(\delta\omega)])$.*

Note that the projection problem (5) is infeasible whenever $\beta < \min\{\boldsymbol{b}\}$. The condition in the statement of Theorem 2 can thus be interpreted as a strict feasibility requirement. It is worth contrasting the result of Theorem 2 with the solution of the projection problem (5) as an exponential cone program. The latter would result in a *practical* complexity of $\mathcal{O}(S^3)$, assuming that—which is often observed in practice—the number of iterations of the employed interior-point solver does not grow with the problem dimensions. A *theoretically guaranteed* complexity, on the other hand, does not seem to be available at present as the commercial state-of-the-art solvers for exponential conic programs are not proven to terminate in polynomial time.

**Corollary 1.** *The robust Bellman iteration* (3) *over a KL divergence ambiguity set can be computed to any accuracy $\epsilon > 0$ in time $\mathcal{O}(S^2 \cdot A \log A \cdot \log[\overline{R}^2 \cdot \log(\min\{\overline{\boldsymbol{p}}\}^{-1})/(\epsilon^2 \kappa)] \cdot \log[\overline{R}/\epsilon])$.*

[14] propose a first-order framework for RMDPs over $s$-rectangular KL divergence ambiguity sets whose robust Bellman update enjoys a complexity of $\mathcal{O}(\ell^2 \cdot S^2 \cdot A \cdot \log(\epsilon^{-1}))$, where $\ell$ is the iteration number. A careful analysis results in an overall convergence rate for the optimal MDP policy of $\mathcal{O}(S^3 \cdot A^2 \cdot \epsilon^{-1} \log[\epsilon^{-1}])$. In contrast, the convergence rate of our robust value iteration amounts to $\mathcal{O}(S^2 \cdot A \log A \cdot \log[\overline{R}^2 \cdot \log(\min\{\overline{\boldsymbol{p}}\}^{-1})/(\epsilon^2 \kappa)] \cdot \log[\overline{R}/\epsilon] \cdot \log[\epsilon^{-1}])$. Treating the problem parameters $\overline{R}$, $\overline{\boldsymbol{p}}$ and $\kappa$ as constants, our convergence rate simplifies to $\mathcal{O}(S^2 \cdot A \log A \cdot \log[\epsilon^{-2}] \cdot \log^2[\epsilon^{-1}])$, which compares favourably against the convergence rate of the first-order scheme. Our numerical results in Section 6 show that this theoretical difference appears to carry over to a favourable empirical performance on test instances as well.

We finally note the related work [1], which optimizes a linear function over the intersection of a probability simplex with a constraint on the KL divergence to a nominal distribution. While one could in principle modify that algorithm to solve our projection problem (5), the resulting algorithm would require an additional bisection and would thus be significantly slower than ours.

## 5.2 Burg Entropy

Similar to the KL divergence, the reduced projection problem (6) can be further simplified to a univariate convex optimization problem for the Burg entropy $\phi(t) = -\log t + t - 1$.

**Proposition 3.** *For the Burg entropy $\phi(t) = -\log t + t - 1$, if $\beta > \min\{\boldsymbol{b}\}$, then the optimal value of the projection problem* (5) *equals the optimal value of the univariate convex problem*

$$\underset{\alpha \in [0,1]}{\text{maximize}} \quad \sum_{s' \in \mathcal{S}} \overline{p}_{sas'} \cdot \log\left(1 + \alpha \frac{b_{s'} - \beta}{\beta - \min\{\boldsymbol{b}\}}\right). \tag{8}$$

Similar to the KL divergence, the univariate optimization problem (8) can be solved efficiently.

**Theorem 3.** *If $\beta \geq \min\{\boldsymbol{b}\} + \omega$ for some $\omega > 0$, then the projection problem* (5) *can be solved to any $\delta$-accuracy in time $\mathcal{O}(S \cdot \log[\max\{\boldsymbol{b}\}/(\delta\omega)])$.*

As with the KL divergence, the projection problem (5) corresponding to the Burg entropy can be solved in a practical complexity of $\mathcal{O}(S^3)$ as an exponential cone program, whereas we are not aware of any state-of-the-art solvers equipped with theoretical guarantees. To our best knowledge, RMDPs with $s$-rectangular Burg entropy ambiguity sets have not been studied previously in the literature.

**Corollary 2.** *The robust Bellman iteration* (3) *over a Burg entropy ambiguity set can be computed to any accuracy $\epsilon > 0$ in time $\mathcal{O}(S^2 \cdot A \log A \cdot \log[\overline{R}^2/(\epsilon^2\kappa)] \cdot \log[\overline{R}/\epsilon])$.*

Similar to the previous subsection, we note that the related paper [1] optimizes a linear function over the intersection of a probability simplex with a bound on the Burg entropy to a nominal distribution. While that algorithm could in principle be employed to solve our projection problem (5), the resulting solution scheme would not be competitive due to the inclusion of an additional bisection.

## 5.3 Variation Distance

We first provide an equivalent univariate optimization problem for the reduced projection problem (6) corresponding to the variation distance $\phi(t) = |t - 1|$.

**Proposition 4.** *For the variation distance $\phi(t) = |t - 1|$, the optimal value of the projection problem* (5) *equals the optimal value of the univariate convex problem*

$$\underset{\alpha \in \mathbb{R}_+}{\text{maximize}} \quad 2 + \alpha(\min\{\boldsymbol{b}\} - \beta) - \sum_{s' \in \mathcal{S}} \overline{p}_{sas'} \cdot [2 + \alpha \cdot (\min\{\boldsymbol{b}\} - b_{s'})]_+. \tag{9}$$

Once more, the univariate optimization problem (9) admits an efficient solution.

**Theorem 4.** *The projection problem* (5) *can be solved exactly in time $\mathcal{O}(S \log S)$.*

Note that in contrast to the previous results, Theorem 4 employs a binary search and thus offers an *exact* solution to the projection problem (5). Our result of Theorem 4 matches the complexity of the homotopy continuation method proposed by [19]. The correctness and runtime of their algorithm, however, relies on lengthy ad hoc arguments, whereas Theorem 4 relies on the groundwork laid by Theorem 1 and Proposition 1. Problem (5) can also be solved as a linear program with a practical complexity of $\mathcal{O}(S^3)$ and a theoretical complexity of $\mathcal{O}(S^{3.5})$.

**Corollary 3.** *The robust Bellman iteration* (3) *over a variation distance ambiguity set can be computed to any accuracy $\epsilon > 0$ in time $\mathcal{O}(S^2 \log S \cdot A \cdot \log[\overline{R}/\epsilon])$.*

[40] study the related problem of optimizing a linear function over the intersection of a probability simplex with an unweighted 1-norm constraint, and they identify structural properties of the optimal solutions. Since the linear function and the norm constraint are in different places of the optimization problem, however, their findings are not directly applicable to our setting.

## 5.4 $\chi^2$-Distance

In contrast to the previous subsections, we directly solve the bivariate problem (6) for the $\chi^2$-distance $\phi(t) = (t - 1)^2$ without first formulating an associated univariate optimization problem.

**Theorem 5.** *For the $\chi^2$-distance $\phi(t) = (t - 1)^2$, the optimal value of the projection problem* (5) *can be computed exactly in time $\mathcal{O}(S \log S)$.*

Theorem 5 splits the bivariate piecewise quadratic optimization problem (6) corresponding to the $\chi^2$-distance into $S + 1$ bivariate quadratic problems by sorting the components of $\boldsymbol{b}$. Each of these $S + 1$ problems can be reduced to the solution of 3 univariate quadratic problems that themselves admit analytical solutions.

| $S$ | MOSEK | fast | MOSEK/fast |
|---|---|---|---|
| 1,000 | 47.56 | 0.20 | 243.23 |
| 1,500 | 71.00 | 0.29 | 241.92 |
| 2,000 | 89.57 | 0.39 | 231.46 |
| 2,500 | 114.82 | 0.48 | 239.11 |
| 3,000 | 139.13 | 0.58 | 241.86 |

| $S = A$ | MOSEK | fast | MOSEK/fast |
|---|---|---|---|
| 100 | 3,383.35 | 22.32 | 151.56 |
| 150 | 15,353.40 | 51.67 | 297.17 |
| 200 | 46,049.30 | 83.86 | 549.10 |
| 250 | 104,709.00 | 130.34 | 803.35 |
| 300 | 215,871.00 | 176.35 | 1,224.05 |

Table 2: Comparison of our algorithms ('fast') vs. MOSEK for the projection problem (left) and the Bellman update (right) on KL-divergence constrained ambiguity sets. Runtimes are reported in ms.

| $S = A$ | f-o (3 its) | f-o (5 its) | fast | f-o/fast (3 its) | f-o/fast (5 its) |
|---|---|---|---|---|---|
| 100 | 175.80 | 489.66 | 22.32 | 7.88 | 21.94 |
| 150 | 397.84 | 1,103.53 | 51.67 | 7.70 | 21.36 |
| 200 | 704.93 | 1,955.05 | 83.86 | 8.41 | 23.31 |
| 250 | 1,051.25 | 2,921.71 | 130.34 | 8.07 | 22.42 |
| 300 | 1,542.03 | 4,283.27 | 176.36 | 8.74 | 24.29 |

Table 3: Comparison of our algorithms ('fast') vs. the first-order method of [14] (after $\ell = 3, 5$ its.) for the Bellman update on KL-divergence constrained ambiguity sets. Runtimes are reported in ms.

**Corollary 4.** *The robust Bellman iteration* (3) *over a $\chi^2$-distance ambiguity set can be computed to any accuracy $\epsilon > 0$ in time $\mathcal{O}(S^2 \log S \cdot A \cdot \log[\overline{R}/\epsilon])$.*

The projection problem (5) for the $\chi^2$-distance ambiguity set can be solved as a quadratic program with a practical complexity of $\mathcal{O}(S^3)$ as well as a theoretical complexity of $\mathcal{O}(S^{3.5})$.

The first-order framework of [14] also applies to RMDPs over $s$-rectangular spherical uncertainty sets. In that case, the robust Bellman update enjoys a complexity of $\mathcal{O}(\ell^2 \cdot S^2 \cdot A \cdot \log^2(\epsilon^{-1}))$, where $\ell$ is the iteration number. A careful analysis results in an overall convergence rate for the optimal MDP policy of $\mathcal{O}(S^3 \log S \cdot A^2 \cdot \epsilon^{-1} \log[\epsilon^{-1}])$. In contrast, the convergence rate of our robust value iteration amounts to $\mathcal{O}(S^2 \log S \cdot A \cdot \log[\overline{R}/\epsilon] \cdot \log[\epsilon^{-1}])$. Treating the parameter $\overline{R}$ as a constant, our convergence rate simplifies to $\mathcal{O}(S^2 \log S \cdot A \cdot \log^2[\epsilon^{-1}])$, which compares favourably against the convergence rate of [14]. We remark, however, that the spherical ambiguity sets of [14] differ from the $\chi^2$-distance ambiguity sets studied here, and as such the two methods are not directly comparable. We also note that our $\chi^2$-distance ambiguity sets enjoy a strong statistical justification [4, 6].

Computing unweighted 2-norm projections of points onto $S$-dimensional probability simplices has manifold applications in image processing, finance, optimization and machine learning [1, 8]. [30] proposes one of the earliest algorithms that computes this projection in time $\mathcal{O}(S^2)$ by iteratively reducing the dimension of the problem using Lagrange multipliers. The minimum complexity of $\mathcal{O}(S)$ is achieved, among others, by [27] through a linear-time median-finding algorithm and by [36] through a filtered bucket-clustering method. Note, however, that these algorithms do *not* account for the weights and the additional inequality constraint present in our generalized projection (5). The unweighted 2-norm projection of a point onto the intersection of the $S$-dimensional probability simplex with an axis-parallel hypercube is computed by [52] through a sorting-based method and by [2] through Newton's method, respectively. [38] optimize a linear function over the intersection of a probability simplex with an unweighted 2-norm constraint through an iterative dimension reduction scheme. [1], finally, study algorithms that optimize linear functions over the intersection of a probability simplex and a bound on the unweighted 2-norm distance to a nominal distribution.

## 6 Numerical Results

We compare our fast suite of algorithms with the state-of-the-art solver MOSEK 9.3 [3] (commercial) and the first-order method of [14]. All experiments are implemented in C++, and they are run on a 3.6 GHz 8-Core Intel Core i9 CPU with 32 GB 2667 MHz DDR4 main memory. The source code is available at https://sites.google.com/view/clint-chin-pang-ho.

| $S$ | MOSEK | fast | MOSEK/fast | $S = A$ | MOSEK | fast | MOSEK/fast |
|---|---|---|---|---|---|---|---|
| 1,000 | 49.94 | 0.05 | 981.91 | 100 | 148.63 | 2.59 | 57.40 |
| 1,500 | 76.61 | 0.08 | 945.99 | 150 | 353.58 | 6.01 | 58.85 |
| 2,000 | 93.13 | 0.11 | 854.39 | 200 | 687.14 | 11.78 | 58.34 |
| 2,500 | 123.16 | 0.14 | 879.72 | 250 | 1,212.30 | 19.38 | 62.54 |
| 3,000 | 153.06 | 0.17 | 917.18 | 300 | 1,908.92 | 25.84 | 73.87 |

Table 4: Comparison of our algorithms ('fast') vs. MOSEK for the projection problem (left) and the Bellman update (right) on $\chi^2$-distance constrained ambiguity sets. Runtimes are reported in ms.

For our experiments, we synthetically generate random RMDP instances as follows. For the projection problem, we sample each component of $b$ uniformly at random between 0 and 1. Similarly, we sample each component of $\overline{p}_{sa}$ uniformly at random between 0 and 1 and subsequently scale $\overline{p}_{sa}$ so that its elements sum up to 1. The parameter $\beta$, finally, is uniformly distributed between $\min\{b\} + 10^{-8}$ and $\overline{p}_{sa}^\top b - 10^{-8}$ to adhere to the assumptions of our paper. For the robust Bellman update, all vectors $b_{sa}$ and all transition probabilities $p_{sa}$, $s \in \mathcal{S}$ and $a \in \mathcal{A}$, are generated according to the above procedure. The parameter $\kappa$ is also sampled from a uniform distribution supported on $[0, 1]$.

Tables 2–4 report average computation times over 50 randomly generated test instances for the KL-divergence and the $\chi^2$-distance based ambiguity sets and show that the proposed algorithms outperform other methods. The tables reveal that our algorithms are about two orders of magnitude faster than MOSEK in solving the projection problem (5). For computing the robust Bellman update $\mathfrak{J}$, our algorithms are also orders of magnitude faster than MOSEK, and this ratio increases with the problem size. The results also show that our algorithm outperforms the first-order method of [14]. The difference becomes more significant with an increasing number of robust Bellman updates. While our algorithms outperform the first-order scheme by a factor of 8 in the third Bellman iteration, they outperform it by a factor of about 20 in the fifth Bellman iteration. Since first-order methods are known to require many iterations for convergence, we conclude that our algorithm compares favorably in this experiment as well.

## 7   Conclusion

We study $s$-rectangular robust MDPs with $\phi$-divergence ambiguity sets. We develop efficient algorithms for computing robust Bellman updates for several important special cases of this ambiguity set. Our experimental results indicate that the proposed algorithms outperform MOSEK. Future work should address extensions to scalable model-free algorithms.

### Acknowledgments

This work was supported, in part, by the Engineering and Physical Sciences Research Council (EPSRC) grant EP/W003317/1, by the CityU Start-Up Grant (Project No. 9610481), by the National Natural Science Foundation of China (Project No. 72032005), by the Chow Sang Sang Group Research Fund sponsored by Chow Sang Sang Holdings International Limited (Project No. 9229076), and by the NSF grant No. 1815275. Any opinion, finding, conclusion, or recommendation expressed in this material are those of the authors and do not necessarily reflect the views of the Engineering and Physical Sciences Research Council and the National Natural Science Foundation of China.

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
