# A   Additional Results of the Experiments

Figures 2–4 provide error bars (with mean values as horizontal curves and $\pm$ standard deviations as vertical bars) for the numerical results reported in Tables 2–4.

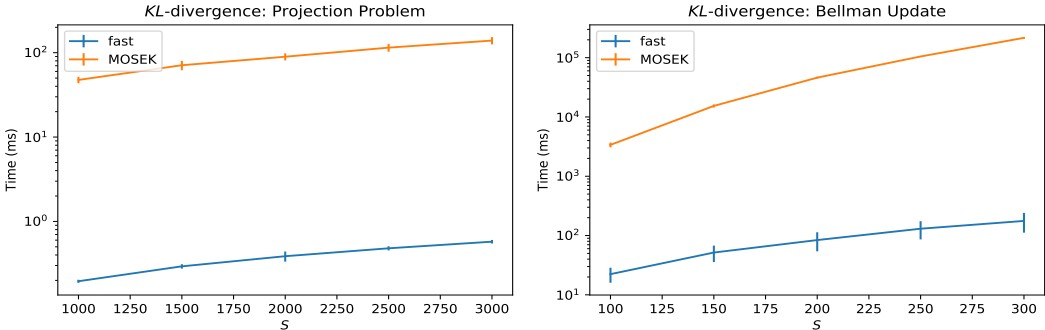

Figure 2: Comparison of our algorithms ('fast') vs. MOSEK on KL-divergence constrained ambiguity sets, as shown in Table 2. *Left*: projection problem. *Right*: Bellman update.

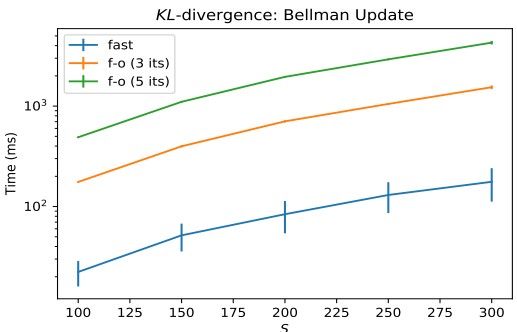

Figure 3: Comparison of our algorithms ('fast') vs. the first-order method of [14] (after $\ell = 3, 5$ its.) for the Bellman update on KL-divergence constrained ambiguity sets, as shown in Table 3.

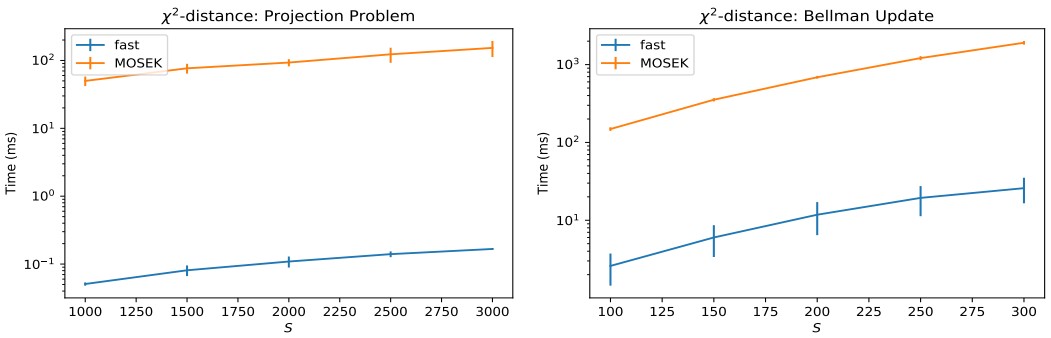

Figure 4: Comparison of our algorithms ('fast') vs. MOSEK on $\chi^2$-distance constrained ambiguity sets, as shown in Table 4. *Left*: projection problem. *Right*: Bellman update.

# B Proofs

The proof of Theorem 1 requires us to analyze the quantitative stability of the robust Bellman operator $\mathfrak{J}(\boldsymbol{v})$. To this end, we study the dual of problem (3') on page 17, which the proof of Theorem 1 will show to be equivalent to the robust value iteration (3):

$$
\begin{aligned}
\text{maximize} \quad & -\kappa\omega + \mathbf{e}^\top\boldsymbol{\gamma} - \omega\sum_{a\in\mathcal{A}} d_a^\star\left(\frac{1}{\omega}\left[\boldsymbol{\theta}_a + \gamma_a\mathbf{e} - \alpha_a(\boldsymbol{r}_{sa}+\lambda\boldsymbol{v})\right], \overline{\boldsymbol{p}}_{sa}\right) \\
\text{subject to} \quad & \boldsymbol{\alpha}\in\Delta_A, \ \ \omega\in\mathbb{R}_+, \ \ \boldsymbol{\gamma}\in\mathbb{R}^A, \ \ \boldsymbol{\theta}\in\mathbb{R}_+^{AS}
\end{aligned}
\tag{10}
$$

Here, $d_a^\star(\boldsymbol{x},\overline{\boldsymbol{p}}_{sa}) := \sup\{\boldsymbol{p}_{sa}^\top\boldsymbol{x} - d_a(\boldsymbol{p}_{sa},\overline{\boldsymbol{p}}_{sa}) : \boldsymbol{p}_{sa}\in\mathbb{R}^S\}$ denotes the conjugate of the deviation function $d_a(\cdot,\overline{\boldsymbol{p}}_{sa})$ in the definition (2) of the ambiguity set $\mathcal{P}_s$, and the perspective function in (10) extends to $\omega = 0$ in the usual way [41, Corollary 8.5.2]. Note also that strong duality holds between (3') and (10) since problem (3') affords a Slater point by assumption (**K**).

The proof of Theorem 1 relies on two auxiliary results, which we state and prove first.

**Lemma 1.** *For any primal-dual pair $\boldsymbol{p}_s^\star\in\mathbb{R}^{AS}$ and $(\boldsymbol{\alpha}^\star,\omega^\star,\boldsymbol{\gamma}^\star,\boldsymbol{\theta}^\star)\in\mathbb{R}^A\times\mathbb{R}\times\mathbb{R}^A\times\mathbb{R}^{AS}$ satisfying the Karush-Kuhn-Tucker conditions for the problems (3') and (10), we have that*

$$
\omega^\star \leq \frac{\max\limits_{a\in\mathcal{A}}\|\boldsymbol{r}_{sa}+\lambda\boldsymbol{v}\|_\infty}{\sum\limits_{a\in\mathcal{A}} d_a(\boldsymbol{p}_{sa}^\star,\overline{\boldsymbol{p}}_{sa})},
$$

*where the right-hand side is interpreted as $+\infty$ whenever the denominator is zero.*

**Proof of Lemma 1.** Using the notational shorthand $\boldsymbol{b}_{sa} = \boldsymbol{r}_{sa} + \lambda\boldsymbol{v}$, the KKT conditions for (3') and (10) are:

$$
\begin{aligned}
&\alpha_a\boldsymbol{b}_{sa} - \gamma_a\mathbf{e} - \boldsymbol{\theta}_a + \omega\nabla_{\boldsymbol{p}_{sa}}d_a(\boldsymbol{p}_{sa},\overline{\boldsymbol{p}}_{sa}) = \mathbf{0} \ \ \forall a\in\mathcal{A} && \text{(Stationarity)} \\
&\mathbf{e}^\top\boldsymbol{\alpha} = 1 && \text{(Stationarity)} \\
&\sum_{a\in\mathcal{A}} d_a(\boldsymbol{p}_{sa},\overline{\boldsymbol{p}}_{sa}) \leq \kappa, \ \ \boldsymbol{p}_{sa}^\top\boldsymbol{b}_{sa} \leq B \ \ \forall a\in\mathcal{A}, \ \ \boldsymbol{p}_s\in(\Delta_S)^A, \ \ B\in\mathbb{R} && \text{(Primal Feasibility)} \\
&\boldsymbol{\alpha}\in\mathbb{R}_{++}^A, \ \ \omega\in\mathbb{R}_+, \ \ \boldsymbol{\gamma}\in\mathbb{R}^A, \ \ \boldsymbol{\theta}\in\mathbb{R}_+^{AS} && \text{(Dual Feasibility)} \\
&\alpha_a(\boldsymbol{p}_{sa}^\top\boldsymbol{b}_{sa} - B) = 0 \ \ \forall a\in\mathcal{A} && \text{(Complementary Slackness)} \\
&\omega\left(\sum_{a\in\mathcal{A}} d_a(\boldsymbol{p}_{sa},\overline{\boldsymbol{p}}_{sa}) - \kappa\right) = 0 && \text{(Complementary Slackness)} \\
&\theta_{as'}p_{sas'} = 0 \ \ \forall a\in\mathcal{A}, \ s'\in\mathcal{S} && \text{(Complementary Slackness)}
\end{aligned}
$$

Here, $B\in\mathbb{R}$ denotes the epigraphical variable used to linearize the objective function in (3'). The proof is split into two parts. We first show that for every $a\in\mathcal{A}$ there is $s'\in\mathcal{S}$ such that

$$
d_a(\boldsymbol{p}_{sa},\overline{\boldsymbol{p}}_{sa}) \leq \boldsymbol{p}_{sa}^\top\nabla_{\boldsymbol{p}_{sa}}d_a(\boldsymbol{p}_{sa},\overline{\boldsymbol{p}}_{sa}) - [\nabla_{\boldsymbol{p}_{sa}}d_a(\boldsymbol{p}_{sa},\overline{\boldsymbol{p}}_{sa})]_{s'}.
\tag{11}
$$

We next prove that for all $s'\in\mathcal{S}$ and $a\in\mathcal{A}$, we have

$$
\omega\left(\boldsymbol{p}_{sa}^\top\nabla_{\boldsymbol{p}_{sa}}d_a(\boldsymbol{p}_{sa},\overline{\boldsymbol{p}}_{sa}) - [\nabla_{\boldsymbol{p}_{sa}}d_a(\boldsymbol{p}_{sa},\overline{\boldsymbol{p}}_{sa})]_{s'}\right) \leq \alpha_a b_{sas'}.
\tag{12}
$$

Since $\omega\in\mathbb{R}_+$ by the dual feasibility condition, (11) and (12) imply that for every $a\in\mathcal{A}$ there is $s'\in\mathcal{S}$ such that $\omega d_a(\boldsymbol{p}_{sa},\overline{\boldsymbol{p}}_{sa}) \leq \alpha_a b_{sas'}$. From this we obtain that

$$
\omega\sum_{a\in\mathcal{A}} d_a(\boldsymbol{p}_{sa},\overline{\boldsymbol{p}}_{sa}) \leq \sum_{a\in\mathcal{A}}\alpha_a\max_{s'\in\mathcal{S}}\{b_{sas'}\} \leq \max_{a\in\mathcal{A},s'\in\mathcal{S}}\{b_{sas'}\},
$$

where the last inequality holds since $\mathbf{e}^\top\boldsymbol{\alpha} = 1$ by the second stationarity condition. This proves the statement of the lemma.

To show (11), we note that

$$
d_a(\boldsymbol{p}_{sa},\overline{\boldsymbol{p}}_{sa}) + \nabla_{\boldsymbol{p}_{sa}}d_a(\boldsymbol{p}_{sa},\overline{\boldsymbol{p}}_{sa})^\top(\overline{\boldsymbol{p}}_{sa} - \boldsymbol{p}_{sa}) \leq d_a(\overline{\boldsymbol{p}}_{sa},\overline{\boldsymbol{p}}_{sa}) = 0
$$

since $d_a$ is convex by assumption (**C**) and $d_a(\overline{\boldsymbol{p}}_{sa},\overline{\boldsymbol{p}}_{sa}) = 0$ by assumption (**D**). We thus have

$$
d_a(\boldsymbol{p}_{sa},\overline{\boldsymbol{p}}_{sa}) \leq \boldsymbol{p}_{sa}^\top\nabla_{\boldsymbol{p}_{sa}}d_a(\boldsymbol{p}_{sa},\overline{\boldsymbol{p}}_{sa}) - \overline{\boldsymbol{p}}_{sa}^\top\nabla_{\boldsymbol{p}_{sa}}d_a(\boldsymbol{p}_{sa},\overline{\boldsymbol{p}}_{sa}),
$$

and the fact that $\overline{\boldsymbol{p}}_{sa} \in \Delta_S$ implies that

$$d_a(\boldsymbol{p}_{sa}, \overline{\boldsymbol{p}}_{sa}) \leq \boldsymbol{p}_{sa}^\top \nabla_{\boldsymbol{p}_{sa}} d_a(\boldsymbol{p}_{sa}, \overline{\boldsymbol{p}}_{sa}) - \min_{s' \in \mathcal{S}} \, [\nabla_{\boldsymbol{p}_{sa}} d_a(\boldsymbol{p}_{sa}, \overline{\boldsymbol{p}}_{sa})]_{s'} \,,$$

which is equivalent to (11).

We now prove (12). Aggregating the equations in the first stationarity condition according to the weights $\boldsymbol{p}_{sa} \in \Delta_S$ shows that for all $a \in \mathcal{A}$, we have

$$\alpha_a \boldsymbol{p}_{sa}^\top \boldsymbol{b}_{sa} - \gamma_a \mathbf{e}^\top \boldsymbol{p}_{sa} - \boldsymbol{p}_{sa}^\top \boldsymbol{\theta}_a + \omega \boldsymbol{p}_{sa}^\top \nabla_{\boldsymbol{p}_{sa}} d_a(\boldsymbol{p}_{sa}, \overline{\boldsymbol{p}}_{sa}) = 0$$

$$\Longleftrightarrow \quad \gamma_a = \alpha_a \boldsymbol{p}_{sa}^\top \boldsymbol{b}_{sa} + \omega \boldsymbol{p}_{sa}^\top \nabla_{\boldsymbol{p}_{sa}} d_a(\boldsymbol{p}_{sa}, \overline{\boldsymbol{p}}_{sa}) \tag{13}$$

since the primal feasibility condition guarantees that $\mathbf{e}^\top \boldsymbol{p}_{sa} = 1$ and the last complementary slackness condition ensures that $\boldsymbol{p}_{sa}^\top \boldsymbol{\theta}_a = 0$. However, the first stationarity condition also implies

$$\alpha_a b_{sas'} - \gamma_a - \theta_{as'} + \omega \, [\nabla_{\boldsymbol{p}_{sa}} d_a(\boldsymbol{p}_{sa}, \overline{\boldsymbol{p}}_{sa})]_{s'} = 0 \quad \forall a \in \mathcal{A}, \, s' \in \mathcal{S}$$

$$\Longleftrightarrow \quad \gamma_a \leq \alpha_a b_{sas'} + \omega \, [\nabla_{\boldsymbol{p}_{sa}} d_a(\boldsymbol{p}_{sa}, \overline{\boldsymbol{p}}_{sa})]_{s'} \quad \quad \forall a \in \mathcal{A}, \, s' \in \mathcal{S} \tag{14}$$

since $\theta_{as'} \geq 0$ due to the dual feasibility condition. Combining (13) and (14), finally, yields

$$\alpha_a \boldsymbol{p}_{sa}^\top \boldsymbol{b}_{sa} + \omega \boldsymbol{p}_{sa}^\top \nabla_{\boldsymbol{p}_{sa}} d_a(\boldsymbol{p}_{sa}, \overline{\boldsymbol{p}}_{sa}) \leq \alpha_a b_{sas'} + \omega \, [\nabla_{\boldsymbol{p}_{sa}} d_a(\boldsymbol{p}_{sa}, \overline{\boldsymbol{p}}_{sa})]_{s'} \quad \forall a \in \mathcal{A}, s' \in \mathcal{S}$$

$$\Longleftrightarrow \quad \omega \left( \boldsymbol{p}_{sa}^\top \nabla_{\boldsymbol{p}_{sa}} d_a(\boldsymbol{p}_{sa}, \overline{\boldsymbol{p}}_{sa}) - [\nabla_{\boldsymbol{p}_{sa}} d_a(\boldsymbol{p}_{sa}, \overline{\boldsymbol{p}}_{sa})]_{s'} \right) \leq \alpha_a b_{sas'} - \alpha_a \boldsymbol{p}_{sa}^\top \boldsymbol{b}_{sa} \quad \forall a \in \mathcal{A}, s' \in \mathcal{S},$$

which implies (12) since $\alpha_a \boldsymbol{p}_{sa}^\top \boldsymbol{b}_{sa} \geq 0$ as $\alpha_a \geq 0$ by the dual feasibility condition, $\boldsymbol{p}_{sa} \geq \mathbf{0}$ by the primal feasibility condition and $\boldsymbol{b}_{sa} \geq \mathbf{0}$ by assumption. $\qquad \square$

**Lemma 2.** *Let $\mathfrak{J}(\boldsymbol{v}; \kappa)$ be the robust Bellman iterate (3) with the budget $\kappa > 0$ in the ambiguity set $\mathcal{P}$. For any $\kappa' \geq \kappa$ and any primal-dual pair $\boldsymbol{p}_s^\star \in \mathbb{R}^{AS}$ and $(\boldsymbol{\alpha}^\star, \omega^\star, \boldsymbol{\gamma}^\star, \boldsymbol{\theta}^\star) \in \mathbb{R}^A \times \mathbb{R} \times \mathbb{R}^A \times \mathbb{R}^{AS}$ satisfying the Karush-Kuhn-Tucker conditions for (3') and (10) with budget $\kappa$, we have*

$$\|\mathfrak{J}(\boldsymbol{v}; \kappa) - \mathfrak{J}(\boldsymbol{v}; \kappa')\|_\infty \leq \frac{(\kappa' - \kappa) \max\limits_{a \in \mathcal{A}} \|\boldsymbol{r}_{sa} + \lambda \boldsymbol{v}\|_\infty}{\sum\limits_{a \in \mathcal{A}} d_a(\boldsymbol{p}_{sa}^\star, \overline{\boldsymbol{p}}_{sa})},$$

*where the right-hand side is interpreted as $+\infty$ whenever the denominator is zero.*

**Proof of Lemma 2.** Since $\kappa' \geq \kappa$, we have for fixed $s \in \mathcal{S}$ that

$$|[\mathfrak{J}(\boldsymbol{v}; \kappa)]_s - [\mathfrak{J}(\boldsymbol{v}; \kappa')]_s| = [\mathfrak{J}(\boldsymbol{v}; \kappa)]_s - [\mathfrak{J}(\boldsymbol{v}; \kappa')]_s \leq \omega^\star (\kappa' - \kappa),$$

where $\omega^\star$ belongs to any primal-dual pair $\boldsymbol{p}_s^\star \in \mathbb{R}^{AS}$ and $(\boldsymbol{\alpha}^\star, \omega^\star, \boldsymbol{\gamma}, \boldsymbol{\theta}^\star) \in \mathbb{R}^A \times \mathbb{R} \times \mathbb{R}^A \times \mathbb{R}^{AS}$ satisfying the KKT conditions of problems (3') and (10). Indeed, since only the first term in the objective function in (10) depends on $\kappa$, the solution $(\boldsymbol{\alpha}^\star, \omega^\star, \boldsymbol{\gamma}, \boldsymbol{\theta}^\star)$ for the dual problem with budget $\kappa$ remains feasible (but is typically not optimal) for the dual problem with budget $\kappa'$, and its objective value decreases by precisely $\omega^\star (\kappa' - \kappa)$. The result now follows from Lemma 1. $\qquad \square$

**Proof of Theorem 1.** We compute an $\epsilon$-optimal solution $\boldsymbol{v}'$ to the robust Bellman iteration $\mathfrak{J}(\boldsymbol{v})$ component-wise. To this end, consider any component $v_s'$, $s \in \mathcal{S}$. We apply the classical min-max theorem to equivalently reformulate the right-hand side of (3) as the optimal value of the optimization problem

$$
\begin{aligned}
\text{minimize} \quad & \max_{a \in \mathcal{A}} \left\{ \boldsymbol{p}_{sa}^\top (\boldsymbol{r}_{sa} + \lambda \boldsymbol{v}) \right\} \\
\text{subject to} \quad & \sum_{a \in \mathcal{A}} d_a(\boldsymbol{p}_{sa}, \overline{\boldsymbol{p}}_{sa}) \leq \kappa \\
& \boldsymbol{p}_s \in (\Delta_S)^A.
\end{aligned}
\tag{3'}
$$

In this reformulation, we have replaced the inner maximization over $\boldsymbol{\pi}_s \in \Delta_A$ with the maximization over the extreme points of $\Delta_A$, which is allowed since the objective function is linear in $\boldsymbol{\pi}_s$.

We compute $v_s'$ through a bisection on the optimal value of problem (3'). To this end, we set $\delta = \epsilon \kappa / [2A\overline{R} + A\epsilon]$. We start the bisection with the lower and upper bounds $\underline{v}_s^0 = \underline{R}_s(\boldsymbol{v})$ and $\overline{v}_s^0 = \overline{R}$, respectively. Here, the lower bound satisfies $\underline{R}_s(\boldsymbol{v}) = \max_{a \in \mathcal{A}} \min_{s' \in \mathcal{S}} \{r_{sas'} + \lambda v_{s'}\}$, and it indeed constitutes a lower bound since the projection subproblem (5) is infeasible if $\theta = \beta < \min\{\boldsymbol{b}\}$ which is set to be $\min\{\boldsymbol{r}_{sa} + \lambda \boldsymbol{v}\}$ for each $a \in \mathcal{A}$. Likewise, the upper bound is valid as long as our

robust Bellman iteration operates on (approximate) lower bounds of the value function, which can be guaranteed, for example, by starting the robust Bellman iteration with the initial estimate $v^0 = \mathbf{0}$. In each iteration $i = 0, 1, \ldots$, we consider the midpoint $\theta = (\underline{v}_s^i + \overline{v}_s^i)/2$ and compute the generalized $d_a$-projections $\mathfrak{P}(\overline{\boldsymbol{p}}_{sa}; \boldsymbol{r}_{sa} + \lambda \boldsymbol{v}, \theta)$, $a \in \mathcal{A}$, to $\delta$-accuracy, resulting in the action-wise lower and upper bounds $(\underline{d}_a, \overline{d}_a)$, respectively. When then update the interval bounds as follows:

$$
\begin{cases}
(\underline{v}_s^{i+1}, \overline{v}_s^{i+1}) \leftarrow (\underline{v}_s^i, \theta) & \text{if } \sum_{a \in \mathcal{A}} \overline{d}_a \leq \kappa, \\
(\underline{v}_s^{i+1}, \overline{v}_s^{i+1}) \leftarrow (\theta, \overline{v}_s^i) & \text{if } \sum_{a \in \mathcal{A}} \underline{d}_a > \kappa
\end{cases}
$$

We terminate the bisection once *(i)* $\overline{v}_s^i - \underline{v}_s^i \leq \epsilon$ or *(ii)* $\kappa \in \left[ \sum_{a \in \mathcal{A}} \underline{d}_a, \sum_{a \in \mathcal{A}} \overline{d}_a \right)$, whichever condition holds first. Note that both interval updates ensure that $\underline{v}_s^{i+1}$ and $\overline{v}_s^{i+1}$ remain valid bounds since

$$
\sum_{a \in \mathcal{A}} \overline{d}_a \leq \kappa \quad \Longrightarrow \quad \sum_{a \in \mathcal{A}} \mathfrak{P}(\overline{\boldsymbol{p}}_{sa}; \boldsymbol{r}_{sa} + \lambda \boldsymbol{v}, \theta) \leq \kappa \quad \Longrightarrow \quad [\mathfrak{J}(\boldsymbol{v})]_s \leq \theta
$$

as well as

$$
\sum_{a \in \mathcal{A}} \underline{d}_a > \kappa \quad \Longrightarrow \quad \sum_{a \in \mathcal{A}} \mathfrak{P}(\overline{\boldsymbol{p}}_{sa}; \boldsymbol{r}_{sa} + \lambda \boldsymbol{v}, \theta) > \kappa \quad \Longrightarrow \quad [\mathfrak{J}(\boldsymbol{v})]_s > \theta,
$$

where the respective second implications hold since

$$
\begin{aligned}
& \sum_{a \in \mathcal{A}} \mathfrak{P}(\overline{\boldsymbol{p}}_{sa}; \boldsymbol{r}_{sa} + \lambda \boldsymbol{v}, \theta) \leq \kappa \\
\Longleftrightarrow \quad & \sum_{a \in \mathcal{A}} \min \left\{ d_a(\boldsymbol{p}_{sa}, \overline{\boldsymbol{p}}_{sa}) : \boldsymbol{p}_{sa} \in \Delta_S, \ \boldsymbol{p}_{sa}^\top (\boldsymbol{r}_{sa} + \lambda \boldsymbol{v}) \leq \theta \right\} \leq \kappa \\
\Longleftrightarrow \quad & \exists \boldsymbol{p}_s \in (\Delta_S)^A : \ \sum_{a \in \mathcal{A}} d_a(\boldsymbol{p}_{sa}, \overline{\boldsymbol{p}}_{sa}) \leq \kappa \text{ and } \boldsymbol{p}_{sa}^\top (\boldsymbol{r}_{sa} + \lambda \boldsymbol{v}) \leq \theta \ \forall a \in \mathcal{A},
\end{aligned}
$$

and the latter is the case if and only if the optimal value of problem (3') does not exceed $\theta$, that is, if and only if $[\mathfrak{J}(\boldsymbol{v})]_s \leq \theta$ for some $\theta \in \mathbb{R}$.

At termination, in case *(i)* we have $\overline{v}_s^i - \underline{v}_s^i \leq \epsilon$, which implies that $\theta = (\underline{v}_s^i + \overline{v}_s^i)/2$ is an $\epsilon$-optimal solution to $[\mathfrak{J}(\boldsymbol{v})]_s$. If case *(ii)* is satisfied at termination, on the other hand, then

$$
\sum_{a \in \mathcal{A}} \underline{d}_a \leq \kappa < \sum_{a \in \mathcal{A}} \overline{d}_a \quad \Longrightarrow \quad \sum_{a \in \mathcal{A}} \mathfrak{P}(\overline{\boldsymbol{p}}_{sa}; \boldsymbol{r}_{sa} + \lambda \boldsymbol{v}, \theta) - A\delta \leq \kappa < \sum_{a \in \mathcal{A}} \mathfrak{P}(\overline{\boldsymbol{p}}_{sa}; \boldsymbol{r}_{sa} + \lambda \boldsymbol{v}, \theta) + A\delta,
$$

in which case $\theta = (\underline{v}_s^i + \overline{v}_s^i)/2$ is an *exact* optimal solution to the variant $[\mathfrak{J}(\boldsymbol{v}; \kappa')]_s$ of the robust value iteration (3) where the budget $\kappa$ in the ambiguity set is replaced with some $\kappa' \in [\kappa - A\delta, \ \kappa + A\delta]$. In this case, we have that

$$
\left| \theta - [\mathfrak{J}(\boldsymbol{v})]_s \right| \ \leq \ [\mathfrak{J}(\boldsymbol{v}; \kappa - A\delta)]_s - [\mathfrak{J}(\boldsymbol{v}; \kappa + A\delta)]_s \ \leq \ \epsilon,
$$

where the first inequality follows from the monotonicity of $\mathfrak{J}(\boldsymbol{v}; \cdot)$ in its second argument, and the second inequality holds because of the following argument. If the constraint $\sum_{a \in \mathcal{A}} d_a(\boldsymbol{p}_{sa}, \overline{\boldsymbol{p}}_{sa}) \leq \kappa - A\delta$ in problem (3') is not binding at optimality, then $[\mathfrak{J}(\boldsymbol{v}; \kappa - A\delta)]_s - [\mathfrak{J}(\boldsymbol{v}; \kappa + A\delta)]_s = 0 < \epsilon$. On the other hand, if the constraint $\sum_{a \in \mathcal{A}} d_a(\boldsymbol{p}_{sa}, \overline{\boldsymbol{p}}_{sa}) \leq \kappa - A\delta$ in problem (3') is binding at optimality, then by applying Lemma 2 in the appendix and using our definition of $\delta$ and the fact that $\|\boldsymbol{r}_{sa} + \lambda \boldsymbol{v}\|_\infty \leq \overline{R}$, we have

$$
[\mathfrak{J}(\boldsymbol{v}; \kappa - A\delta)]_s - [\mathfrak{J}(\boldsymbol{v}; \kappa + A\delta)]_s \leq \frac{2A\delta \max_{a \in \mathcal{A}} \|\boldsymbol{r}_{sa} + \lambda \boldsymbol{v}\|_\infty}{\kappa - A\delta} \leq \epsilon.
$$

One readily verifies that at most $\mathcal{O}(\log[\overline{R}/\epsilon])$ iterations of complexity $\mathcal{O}(A \cdot h(\delta))$ are executed in each of the $S$ bisections, which concludes the proof. $\qquad \square$

**Proof of Proposition 1.** For the deviation measure from the statement of this proposition, problem (5) becomes

$$
\begin{aligned}
\text{minimize} \quad & \sum_{s' \in \mathcal{S}} \overline{p}_{sas'} \cdot \phi \left( \frac{p_{sas'}}{\overline{p}_{sas'}} \right) \\
\text{subject to} \quad & \boldsymbol{b}^\top \boldsymbol{p}_{sa} \leq \beta \\
& \boldsymbol{p}_{sa} \in \Delta_S.
\end{aligned} \tag{15}
$$

The Lagrange dual function associated with this problem is

$$g(\alpha, \zeta) = \inf \left\{ \sum_{s' \in \mathcal{S}} \overline{p}_{sas'} \cdot \phi \left( \frac{p_{sas'}}{\overline{p}_{sas'}} \right) + \alpha(\boldsymbol{b}^\top \boldsymbol{p}_{sa} - \beta) + \zeta(1 - \mathbf{e}^\top \boldsymbol{p}_{sa}) \ : \ \boldsymbol{p}_{sa} \in \mathbb{R}_+^S \right\},$$

where $\alpha \in \mathbb{R}_+$ and $\zeta \in \mathbb{R}$. Rearranging terms, we observe that

$$g(\alpha, \zeta) = -\beta\alpha + \zeta - \sum_{s' \in \mathcal{S}} \overline{p}_{sas'} \cdot \sup \left\{ \frac{p_{sas'}}{\overline{p}_{sas'}} \cdot (-\alpha b_{s'} + \zeta) - \phi \left( \frac{p_{sas'}}{\overline{p}_{sas'}} \right) \ : \ p_{sas'} \in \mathbb{R}_+ \right\},$$

and the suprema inside this expression coincide with the convex conjugates $\phi^\star(-\alpha b_{s'} + \zeta)$, $s' \in \mathcal{S}$. The resulting optimization problem (6) is convex since the conjugates are convex. Moreover, since $\min\{\boldsymbol{b}\} \leq \beta$ by assumption, problem (15) affords a feasible solution, and the linearity of the constraints implies that this solution constitutes a Slater point. We thus conclude that strong duality holds between (6) and (15), that is, their optimal objective values indeed coincide. $\qquad\square$

**Proof of Proposition 2.** Plugging the convex conjugate $\phi^\star(y) = e^y - 1$ of the KL divergence into the bivariate optimization problem (6), we obtain

$$\begin{aligned} \text{maximize} \quad & -\beta\alpha + \zeta - \sum_{s' \in \mathcal{S}} \overline{p}_{sas'} \left( e^{-\alpha b_{s'} + \zeta} - 1 \right) \\ \text{subject to} \quad & \alpha \in \mathbb{R}_+, \ \zeta \in \mathbb{R}. \end{aligned}$$

By rearranging terms, the objective function can be expressed as

$$1 - \beta\alpha + \zeta - e^\zeta \left( \sum_{s' \in \mathcal{S}} \overline{p}_{sas'} \cdot e^{-\alpha b_{s'}} \right), \tag{16}$$

and the first-order optimality condition shows that for fixed $\alpha \in \mathbb{R}_+$, the function is maximized by

$$1 - e^{\zeta^\star} \left( \sum_{s' \in \mathcal{S}} \overline{p}_{sas'} \cdot e^{-\alpha b_{s'}} \right) = 0 \iff \zeta^\star = -\log \left( \sum_{s' \in \mathcal{S}} \overline{p}_{sas'} \cdot e^{-\alpha b_{s'}} \right).$$

Substituting $\zeta^\star$ in (16), we obtain problem (7) as postulated. $\qquad\square$

**Proof of Theorem 2.** We prove the statement in three steps. Step 1 shows that the optimal solution $\alpha^\star$ to (7) is lower and upper bounded by $\underline{\alpha}^0 = 0$ and $\overline{\alpha}^0 = \log \left( \frac{1}{\min\{\overline{\boldsymbol{p}}\}} \right) \cdot \frac{1}{\beta - \min\{\boldsymbol{b}\}}$, respectively. Note that $\overline{\alpha}^0$ is finite due to the assumed strict positivity of $\min\{\overline{\boldsymbol{p}}\}$ and $\beta - \min\{\boldsymbol{b}\}$. Step 2 derives a global upper bound on the derivative of $f(\alpha)$, which we henceforth use to denote of the objective function of problem (7). In conjunction with the concavity of $f$, this will allow us to bound the maximum objective function value over any interval $[\underline{\alpha}, \overline{\alpha}] \subseteq \mathbb{R}_+$. Step 3, finally, employs a bisection search to solve (7) to $\delta$-accuracy in the stated complexity.

As for the first step, the validity of the lower bound $\underline{\alpha}^0$ follows directly from the non-negativity constraint in (7). In view of the upper bound $\overline{\alpha}^0$, we note that

$$\begin{aligned} \overline{\alpha}^0 = \log \left( \frac{1}{\min\{\overline{\boldsymbol{p}}\}} \right) \cdot \frac{1}{\beta - \min\{\boldsymbol{b}\}} \quad &\iff \quad \min\{\overline{\boldsymbol{p}}\} \cdot e^{\overline{\alpha}^0(\beta - \min\{\boldsymbol{b}\})} = 1 \\ &\implies \quad \sum_{s' \in \mathcal{S}} \overline{p}_{sas'} \cdot e^{\overline{\alpha}^0(\beta - b_{s'})} \geq 1 \\ &\iff \quad \sum_{s' \in \mathcal{S}} \overline{p}_{sas'} \cdot e^{-\overline{\alpha}^0 b_{s'}} \geq e^{-\beta\overline{\alpha}^0} \\ &\iff \quad \log \left( \sum_{s' \in \mathcal{S}} \overline{p}_{sas'} \cdot e^{-\overline{\alpha}^0 b_{s'}} \right) \geq -\beta\overline{\alpha}^0 \\ &\iff \quad f(\overline{\alpha}^0) \leq 0. \end{aligned}$$

Since $f(0) = 0$ and $f(\overline{\alpha}^0) \leq 0$ while at the same time $\overline{\alpha}^0 > 0$, we conclude from the concavity of $f$ that $\overline{\alpha}^0$ is indeed a valid upper bound on the maximizer of problem (7).

In view of the second step, we observe that

$$f'(\alpha) \;\leq\; f'(0) \;=\; \frac{\sum_{s'\in\mathcal{S}}\overline{p}_{sas'}\cdot b_{s'}}{\sum_{s'\in\mathcal{S}}\overline{p}_{sas'}} - \beta \;\leq\; \overline{p}_{sa}^{\top}b \;\leq\; \max\{b\} \qquad \forall\alpha\in\mathbb{R}_+,$$

where the first inequality follows from the concavity of $f$ and the other two inequalities hold since $\overline{p}_{sa}\in\Delta_S$. The concavity of $f(\alpha)$ then implies that for any $\alpha\in[\underline{\alpha},\overline{\alpha}]\subset\mathbb{R}_+$, we have

$$f(\underline{\alpha}) \;\leq\; f(\alpha) \;\leq\; f(\underline{\alpha}) + f'(\underline{\alpha})\cdot(\overline{\alpha}-\underline{\alpha}) \;\leq\; f(\underline{\alpha}) + \max\{b\}\cdot(\overline{\alpha}-\underline{\alpha}).$$

Thus, if we find $\underline{\alpha}$, $\overline{\alpha}$ sufficiently close such that $\alpha^\star\in[\underline{\alpha},\overline{\alpha}]$, then we can closely bound the optimal objective value of problem (7) from below and above by $f(\underline{\alpha})$ and $f(\underline{\alpha}) + \max\{b\}\cdot(\overline{\alpha}-\underline{\alpha})$, respectively.

As for the third step, finally, we bisect on $\alpha$ by starting with the initial bounds $(\underline{\alpha}^0,\overline{\alpha}^0)$, halving the length of the interval $[\underline{\alpha}^i,\overline{\alpha}^i]$ in each iteration $i=0,1,\dots$ by verifying whether $f'([\underline{\alpha}^i+\overline{\alpha}^i]/2)$ is positive and terminating once $\overline{\alpha}^i-\underline{\alpha}^i\leq\delta/\max\{b\}$. Since $\beta-\min\{b\}\geq\omega$, we have

$$\overline{\alpha}^0 - \underline{\alpha}^0 \;=\; \log\left(\frac{1}{\min\{\overline{p}\}}\right)\cdot\frac{1}{\beta-\min\{b\}} \;\leq\; \frac{1}{\omega}\cdot\log\left(\frac{1}{\min\{\overline{p}\}}\right),$$

and thus the length of the interval no longer exceeds $\delta/\max\{b\}$ once the iteration number $i$ satisfies

$$2^{-i}\cdot(\overline{\alpha}^0-\underline{\alpha}^0) \;\leq\; \frac{\delta}{\max\{b\}} \quad\Longleftarrow\quad 2^{-i}\cdot\frac{1}{\omega}\cdot\log\left(\frac{1}{\min\{\overline{p}\}}\right) \;\leq\; \frac{\delta}{\max\{b\}}$$

$$\Longleftrightarrow\quad i \;\geq\; \log_2\left(\frac{\max\{b\}\cdot\log(\min\{\overline{p}\}^{-1})}{\delta\omega}\right),$$

that is, after $\mathcal{O}(\log[\max\{b\}\cdot\log(\min\{\overline{p}\}^{-1})/(\delta\omega)])$ iterations. The interval $[f(\underline{\alpha}^i),f(\overline{\alpha}^i)]$ then provides the $\delta$-accurate solution to the projection problem (5). The statement now follows from the fact that evaluating the derivative $f'([\underline{\alpha}^i+\overline{\alpha}^i]/2)$ in each bisection step takes time $\mathcal{O}(S)$. $\qquad\square$

**Remark 1.** *The proposed lower and upper bounds in Theorem 2 are tight up to constant factor. Indeed, consider the example where $S=2$ and $A=1$ with $\beta=1.5$, $b=[1\ 2]^\top$ and $\overline{p}=[P\ (1-P)]^\top$ for $P\in(0,0.5)$. Then the objective function in problem (7) is*

$$-1.5\alpha - \log\left(P\cdot\mathrm{e}^{-\alpha}+(1-P)\cdot\mathrm{e}^{-2\alpha}\right).$$

*The above problem satisfies the setting in Theorem 2 since $\min\{b\}=\min\{1,2\}<1.5=\beta$. We search for $\alpha$ that satisfies the first order condition:*

$$-1.5 - \frac{(-1)\cdot P\cdot\mathrm{e}^{-\alpha}+(-2)\cdot(1-P)\cdot\mathrm{e}^{-2\alpha}}{P\cdot\mathrm{e}^{-\alpha}+(1-P)\cdot\mathrm{e}^{-2\alpha}} \;=\; 0$$

$$\Longleftrightarrow\qquad P\cdot\mathrm{e}^{-\alpha}+2\cdot(1-P)\cdot\mathrm{e}^{-2\alpha} \;=\; 1.5\cdot\left(P\cdot\mathrm{e}^{-\alpha}+(1-P)\cdot\mathrm{e}^{-2\alpha}\right)$$

$$\Longleftrightarrow\qquad 0.5\cdot(1-P)\cdot\mathrm{e}^{-2\alpha} \;=\; 0.5\cdot P\cdot\mathrm{e}^{-\alpha}$$

$$\Longleftrightarrow\qquad \frac{1-P}{P} \;=\; \mathrm{e}^{\alpha}$$

$$\Longleftrightarrow\qquad \alpha \;=\; \log\left(\frac{1-P}{P}\right).$$

*We thus have $\alpha^\star=\mathcal{O}(\log(1/P))$, that is, the upper bound on $\alpha^\star$ should be at least $\mathcal{O}(\log(1/P))$, when $P\to0$.*

**Proof of Corollary 1.** The proof of Theorem 1 employs an outer bisection over $\theta$ that requires for each $(s,a)\in\mathcal{S}\times\mathcal{A}$ the repeated solution of the projection problem (7) with $b=r_{sa}+\lambda v$ and $\beta=\theta\in[\underline{R}_s(v)+\frac{\epsilon}{2},\overline{R}-\frac{\epsilon}{2}]$ (since the outer bisection is stopped when the interval length no longer exceeds $\epsilon$) to an accuracy of $\delta=\epsilon\kappa/[2A\overline{R}+A\epsilon]$. In that case, for each $(s,a)\in\mathcal{S}\times\mathcal{A}$ we have

$$\beta-\min\{b\} \;\geq\; \underline{R}_s(v)+\frac{\epsilon}{2}-\min\{r_{sa}+\lambda v\}$$

$$=\; \max_{a\in\mathcal{A}}\min_{s'\in\mathcal{S}}\{r_{sas'}+\lambda v_{s'}\}+\frac{\epsilon}{2}-\min_{a\in\mathcal{A}}\min_{s'\in\mathcal{S}}\{r_{sas'}+\lambda v_{s'}\} \;\geq\; \frac{\epsilon}{2}$$

and $\max\{\boldsymbol{b}\} \leq \overline{R}$. Plugging those estimates into the statement of Theorem 2, we see that the projection problem (7) is solved in time

$$h(\epsilon\kappa/[2A\overline{R} + A\epsilon]) = \mathcal{O}(S \cdot \log[A\overline{R}^2 \cdot \log(\min\{\overline{\boldsymbol{p}}\}^{-1})/(\epsilon^2\kappa)]).$$

Combining this estimate with the complexity $\mathcal{O}(AS \cdot h(\epsilon\kappa/[2A\overline{R} + A\epsilon]) \cdot \log[\overline{R}/\epsilon])$ from Theorem 1, we obtain

$$\mathcal{O}(AS \cdot S \cdot \log[A\overline{R}^2 \cdot \log(\min\{\overline{\boldsymbol{p}}\}^{-1})/(\epsilon^2\kappa)] \cdot \log[\overline{R}/\epsilon]),$$

and a reordering of terms proves the statement of the corollary. $\qquad\square$

**Proof of Proposition 3.** Plugging the convex conjugate $\phi^\star(y) = -\log(1-y)$ of the Burg entropy into the bivariate optimization problem (6), we obtain

$$
\begin{aligned}
\text{maximize} \quad & -\beta\alpha + \zeta + \sum_{s'\in\mathcal{S}} \overline{p}_{sas'} \cdot \log(1 + \alpha b_{s'} - \zeta) \\
\text{subject to} \quad & 1 + \alpha\min\{\boldsymbol{b}\} \geq \zeta \\
& \alpha \in \mathbb{R}_+, \ \zeta \in \mathbb{R}.
\end{aligned}
\tag{17}
$$

Here, the first constraint ensures that the logarithms in the objective function are well-defined (as usual, we assume that $\log 0 = -\infty$). Unlike the proof of Proposition 2, the first-order optimality condition of this problem's objective function does not lend itself to extracting the optimal value of $\zeta$. Instead, we consider the Karush-Kuhn-Tucker conditions for problem (17), which are:

$$
\begin{array}{ll}
\displaystyle\sum_{s'\in\mathcal{S}} \overline{p}_{sas'} \cdot \frac{b_{s'}}{1 + \alpha b_{s'} - \zeta} = \beta - \eta\min\{\boldsymbol{b}\} - \gamma & \text{(Stationarity)} \\[3mm]
\displaystyle\sum_{s'\in\mathcal{S}} \overline{p}_{sas'} \cdot \frac{1}{1 + \alpha b_{s'} - \zeta} = 1 - \eta & \text{(Stationarity)} \\[2mm]
1 + \alpha\min\{\boldsymbol{b}\} - \zeta \geq 0, \ \alpha \in \mathbb{R}_+, \ \zeta \in \mathbb{R} & \text{(Primal Feasibility)} \\
\eta, \gamma \in \mathbb{R}_+ & \text{(Dual Feasibility)} \\
\eta(1 + \alpha\min\{\boldsymbol{b}\} - \zeta) = 0, \ \alpha\gamma = 0 & \text{(Complementary Slackness)}
\end{array}
$$

The optimal value of problem (17) is non-negative since $(\alpha, \zeta) = \boldsymbol{0}$ satisfies the constraints of (17). Hence, complementary slackness implies that $\eta^\star = 0$, as otherwise $1 + \alpha^\star\min\{\boldsymbol{b}\} - \zeta^\star = 0$ would imply that the optimal objective value of problem (17) was $-\infty$. Multiplying the first stationarity condition with $\alpha^\star$ and the second one with $1 - \zeta^\star$ and summing up then yields

$$\alpha^\star\left(\sum_{s'\in\mathcal{S}} \overline{p}_{sas'} \cdot \frac{b_{s'}}{1 + \alpha^\star b_{s'} - \zeta^\star}\right) + (1 - \zeta^\star)\left(\sum_{s'\in\mathcal{S}} \overline{p}_{sas'} \cdot \frac{1}{1 + \alpha^\star b_{s'} - \zeta^\star}\right) = \alpha^\star(\beta - \gamma^\star) + (1 - \zeta^\star)$$

$$\Longleftrightarrow \quad \sum_{s'\in\mathcal{S}} \overline{p}_{sas'} \cdot \frac{1 + \alpha^\star b_{s'} - \zeta^\star}{1 + \alpha^\star b_{s'} - \zeta^\star} = \alpha^\star(\beta - \gamma^\star) + (1 - \zeta^\star) \quad \Longleftrightarrow \quad \zeta^\star = \alpha^\star\beta,$$

where the right-hand side of the first line exploits the fact that $\eta^\star = 0$ and the last equivalence uses complementary slackness to replace $\alpha^\star\gamma^\star$ with 0. The result now follows from substituting $\zeta^\star$ with $\alpha^\star\beta$ in problem (17) and rescaling $\alpha$ via $\alpha \leftarrow (\beta - \min\{\boldsymbol{b}\})\alpha$. $\qquad\square$

**Proof of Theorem 3.** Similar to the proof of Theorem 2, we show the statement in three steps. Step 1 argues that $f(\alpha)$, which we henceforth use to denote of the objective function of problem (8), is well-defined and continuously differentiable on the half-open interval $\alpha \in [0, 1)$ with a positive derivative at 0 and a negative derivative close to 1, respectively. This ensures that the optimum is attained on the open interval $\alpha \in (0, 1)$. Step 2 derives a global upper bound on $f'(\alpha)$, which will allow us to bound the maximum objective function value over any interval $[\underline{a}, \overline{\alpha}] \subseteq \mathbb{R}_+$ due to the concavity of $f$. Step 3, finally, employs a bisection search to solve (8) to $\delta$-accuracy in the stated complexity.

In view of the first step, we note that for $\alpha \in [0, 1)$ we have

$$
\begin{aligned}
(1 - \alpha)(\beta - \min\{\boldsymbol{b}\}) > 0 \quad & \Longleftrightarrow \quad \beta - \min\{\boldsymbol{b}\} + \alpha(\min\{\boldsymbol{b}\} - \beta) > 0 \\
& \Longrightarrow \quad \beta - \min\{\boldsymbol{b}\} + \alpha(b_{s'} - \beta) > 0 \qquad \forall s' \in \mathcal{S} \\
& \Longleftrightarrow \quad 1 + \alpha\frac{b_{s'} - \beta}{\beta - \min\{\boldsymbol{b}\}} > 0 \qquad \forall s' \in \mathcal{S},
\end{aligned}
$$

and thus the expression inside the logarithm of $f(\alpha)$ is strictly positive for all $s' \in \mathcal{S}$. Here, the first inequality holds by assumption, and the last equivalence follows from a division by $\beta - \min\{\boldsymbol{b}\}$, which is strictly positive by assumption. We then observe that for $\alpha \in [0, 1)$, we have

$$f'(\alpha) \;=\; \sum_{s' \in \mathcal{S}} \overline{p}_{sas'} \left[ \left( 1 + \alpha \frac{b_{s'} - \beta}{\beta - \min\{\boldsymbol{b}\}} \right)^{-1} \cdot \frac{b_{s'} - \beta}{\beta - \min\{\boldsymbol{b}\}} \right] \;=\; \sum_{s' \in \mathcal{S}} \overline{p}_{sas'} \left[ \frac{b_{s'} - \beta}{\beta - \min\{\boldsymbol{b}\} + \alpha(b_{s'} - \beta)} \right].$$

In particular, we have $f'(0) = (\overline{\boldsymbol{p}}_{sa}^\top \boldsymbol{b} - \beta)/(\beta - \min\{\boldsymbol{b}\})$, which is positive since $\beta \in \left( \min\{\boldsymbol{b}\}, \overline{\boldsymbol{p}}_{sa}^\top \boldsymbol{b} \right)$ by assumption. (Recall that the projection problem is trivial if $\overline{\boldsymbol{p}}_{sa}^\top \boldsymbol{b} \leq \beta$.) For $\alpha \uparrow 1$, on the other hand, the fractions in $f'(\alpha)$ corresponding to the indices $s' \in \mathcal{S}$ with $b_{s'} = \min\{\boldsymbol{b}\}$ evaluate to $1/(\alpha - 1) \longrightarrow -\infty$, whereas the other fractions evaluate to

$$\frac{b_{s'} - \beta}{(\beta - \min\{\boldsymbol{b}\})(1 - \alpha) + \alpha(b_{s'} - \min\{\boldsymbol{b}\})} \;\longrightarrow\; \frac{b_{s'} - \beta}{\alpha(b_{s'} - \min\{\boldsymbol{b}\})}$$

and thus remain finite. In conclusion, we have $f'(\alpha) < 0$ for $\alpha$ near 1.

As for the second step, we observe that

$$f'(\alpha) \;\leq\; f'(0) \;=\; \frac{\overline{\boldsymbol{p}}_{sa}^\top \boldsymbol{b} - \beta}{\beta - \min\{\boldsymbol{b}\}} \;\leq\; \frac{\max\{\boldsymbol{b}\}}{\beta - \min\{\boldsymbol{b}\}} \;\leq\; \frac{\max\{\boldsymbol{b}\}}{\omega},$$

where the inequalities follow from the concavity of $f$, the fact that $\overline{\boldsymbol{p}}_{sa} \in \Delta_S$ as well as $\beta \geq 0$, and because $\beta - \min\{\boldsymbol{b}\} \geq \omega$, respectively. Similar arguments as in the proof of Theorem 2 then allow us to closely bound the optimal value of problem (8) from below and above by $f(\underline{\alpha})$ and $f(\underline{\alpha}) + f'(\underline{\alpha}) + (\max\{\boldsymbol{b}\}/\omega) \cdot (\overline{\alpha} - \underline{\alpha})$, respectively, whenever $\alpha^\star \in [\underline{\alpha}, \overline{\alpha}]$.

In view of the third step, finally, we bisect on $\alpha$ by starting with the initial bounds $(\underline{\alpha}^0, \overline{\alpha}^0) = (0, 1)$, halving the length of the interval $[\underline{\alpha}^i, \overline{\alpha}^i]$ in each iteration $i = 0, 1, \dots$ by verifying whether $f'([\underline{\alpha}^i + \overline{\alpha}^i]/2)$ is positive and terminating once $\overline{\alpha}^i - \underline{\alpha}^i \leq \delta\omega/\max\{\boldsymbol{b}\}$. Similar arguments as in the proof of Theorem 2 show that this is the case after $\mathcal{O}(\log[\max\{\boldsymbol{b}\}/(\delta\omega)])$ iterations. The statement now follows since evaluating the derivative $f'([\underline{\alpha}^i + \overline{\alpha}^i]/2)$ in each bisection step takes time $\mathcal{O}(S)$. $\qquad\square$

**Proof of Corollary 2.** The proof of Theorem 1 employs an outer bisection over $\theta$ that requires for each $(s, a) \in \mathcal{S} \times \mathcal{A}$ the repeated solution of the projection problem (7) with $\boldsymbol{b} = \boldsymbol{r}_{sa} + \lambda\boldsymbol{v}$ and $\beta = \theta \in [\underline{R}_s(\boldsymbol{v}) + \frac{\epsilon}{2}, \overline{R} - \frac{\epsilon}{2}]$ (since the outer bisection is stopped when the interval length no longer exceeds $\epsilon$) to an accuracy of $\delta = \epsilon\kappa/[2A\overline{R} + A\epsilon]$. In that case, for each $(s, a) \in \mathcal{S} \times \mathcal{A}$ we have $\max\{\boldsymbol{b}\} \leq \overline{R}$. Plugging this estimate into the statement of Theorem 3, we see that the projection problem (8) is solved in time

$$h(\epsilon\kappa/[2A\overline{R} + A\epsilon]) \;=\; \mathcal{O}(S \cdot \log[A\overline{R}^2/(\epsilon^2\kappa)]).$$

Combining this estimate with the complexity $\mathcal{O}(AS \cdot h(\epsilon\kappa/[2A\overline{R} + A\epsilon]) \cdot \log[\overline{R}/\epsilon])$ from Theorem 1, we obtain

$$\mathcal{O}(AS \cdot S \cdot \log[A\overline{R}^2/(\epsilon^2\kappa)] \cdot \log[\overline{R}/\epsilon]),$$

and a reordering of terms proves the statement of the corollary. $\qquad\square$

**Proof of Proposition 4.** Plugging in the definition of $\phi^\star(y)$, see Table 1, results in the following variant of problem (6):

$$\begin{aligned}
\text{maximize} \quad & -\beta\alpha + \zeta - \sum_{s' \in \mathcal{S}} \overline{p}_{sas'} \cdot \max\left\{-1, -\alpha b_{s'} + \zeta\right\} \\
\text{subject to} \quad & \zeta \leq 1 + \alpha b_{s'} \qquad\qquad\qquad\qquad\qquad \forall s' \in \mathcal{S} \\
& \alpha \in \mathbb{R}_+, \;\; \zeta \in \mathbb{R}
\end{aligned}$$

Note that the constraints are equivalent to $\zeta \leq 1 + \alpha\min\{b\}$. The above objective function is piecewise linear in $\zeta$ with coefficients that are all non-negative. Thus, $\zeta^\star = 1 + \alpha\min\{b\}$. In this case, the problem simplifies to the problem in the statement of the proposition. $\qquad\square$

**Proof of Theorem 4.** There must be an optimal solution to this problem that is attained at

$$\alpha^\star \in \{0\} \cup \left\{ \frac{2}{b_{s'} - \min\{\boldsymbol{b}\}} : s' \in \mathcal{S} \right\}.$$

Since the objective function is concave in $\alpha$, we can identify an optimal solution in $\mathcal{O}(\log S)$ iterations via a trisection search. Each evaluation requires time $\mathcal{O}(S)$, thus resulting in an overall complexity of $\mathcal{O}(S \log S)$ as claimed. $\qquad\square$

**Proof of Corollary 3.** Combining the estimate $\mathcal{O}(S \log S)$ from Theorem 4 with the complexity $\mathcal{O}(AS \cdot h(\epsilon\kappa/[2A\overline{R} + A\epsilon]) \cdot \log[\overline{R}/\epsilon])$ from Theorem 1, we obtain

$$\mathcal{O}(AS \cdot S \log S \cdot \log[\overline{R}/\epsilon]),$$

and a reordering of terms proves the statement of the corollary. $\qquad\square$

**Proof of Theorem 5.** To solve this problem, we sort the components of $\boldsymbol{b}$ so that the associated expressions $\{-\alpha b_s + \zeta\}_{s=1}^S$ are monotonically non-decreasing (that is, $\{b_s\}_{s=1}^S$ are monotonically non-increasing). For each $\hat{S} = 0, 1, 2, \ldots, S$, we can then consider the subproblem

$$\begin{aligned}
\text{maximize} \quad & -\beta\alpha + \zeta + \sum_{s'=1}^{\hat{S}} \overline{p}_{sas'} - \sum_{s'=\hat{S}+1}^{S} \overline{p}_{sas'} \cdot \left( (-\alpha b_{s'} + \zeta) + \frac{(-\alpha b_{s'} + \zeta)^2}{4} \right) \\
\text{subject to} \quad & -\alpha b_{s'} + \zeta \leq -2 \qquad \forall s' = 0, \ldots \hat{S} \\
& -\alpha b_{s'} + \zeta \geq -2 \qquad \forall s' = \hat{S}+1, \ldots S \\
& \alpha \in \mathbb{R}_+, \quad \zeta \in \mathbb{R},
\end{aligned}$$

which due to the monotonically non-decreasing ordering of $\{-\alpha b_s + \zeta\}_{s=1}^S$ is equivalent to

$$\begin{aligned}
\text{maximize} \quad & -\beta\alpha + \zeta + \sum_{s'=1}^{\hat{S}} \overline{p}_{sas'} - \sum_{s'=\hat{S}+1}^{S} \overline{p}_{sas'} \cdot \left( (-\alpha b_{s'} + \zeta) + \frac{(-\alpha b_{s'} + \zeta)^2}{4} \right) \\
\text{subject to} \quad & -\alpha b_{\hat{S}} + \zeta \leq -2 \\
& -\alpha b_{\hat{S}+1} + \zeta \geq -2 \\
& \alpha \in \mathbb{R}_+, \quad \zeta \in \mathbb{R}.
\end{aligned} \tag{18}$$

The objective function of (18) can be re-expressed as

$$\left( \sum_{s'=1}^{\hat{S}} \overline{p}_{sas'} \right) - \beta\alpha + \zeta + \left( \sum_{s'=\hat{S}+1}^{S} b_{s'} \overline{p}_{sas'} \right) \alpha - \left( \sum_{s'=\hat{S}+1}^{S} \overline{p}_{sas'} \right) \zeta - \left( \frac{1}{4} \sum_{s'=\hat{S}+1}^{S} b_{s'}^2 \overline{p}_{sas'} \right) \alpha^2$$

$$+ \left( \frac{1}{2} \sum_{s'=\hat{S}+1}^{S} b_{s'} \overline{p}_{sas'} \right) \alpha\zeta - \left( \frac{1}{4} \sum_{s'=\hat{S}+1}^{S} \overline{p}_{sas'} \right) \zeta^2.$$

Recall that for a fixed value of $\alpha$, the constraints of (18) impose that $-2 + \alpha b_{\hat{S}+1} \leq \zeta \leq -2 + \alpha b_{\hat{S}}$. Therefore, for any fixed $\alpha$, problem (18) reduces to a convex quadratic optimization problem with one-dimensional box constraints. One can readily verify that this problem is solved by

$$\zeta^\star = \begin{cases}
-2 + \alpha b_{\hat{S}+1} & \text{if } \dfrac{2\sum_{s'=1}^{\hat{S}} \overline{p}_{sas'} + \left( \sum_{s'=\hat{S}+1}^{S} b_{s'} \overline{p}_{sas'} \right) \alpha}{\left( \sum_{s'=\hat{S}+1}^{S} \overline{p}_{sas'} \right)} \leq -2 + \alpha b_{\hat{S}+1} \\[4ex]
-2 + \alpha b_{\hat{S}} & \text{if } -2 + \alpha b_{\hat{S}} \leq \dfrac{2\sum_{s'=1}^{\hat{S}} \overline{p}_{sas'} + \left( \sum_{s'=\hat{S}+1}^{S} b_{s'} \overline{p}_{sas'} \right) \alpha}{\left( \sum_{s'=\hat{S}+1}^{S} \overline{p}_{sas'} \right)} \\[4ex]
\dfrac{2\sum_{s'=1}^{\hat{S}} \overline{p}_{sas'} + \left( \sum_{s'=\hat{S}+1}^{S} b_{s'} \overline{p}_{sas'} \right) \alpha}{\left( \sum_{s'=\hat{S}+1}^{S} \overline{p}_{sas'} \right)} & \text{otherwise.}
\end{cases}$$

We can consider each case separately. In the first case, we restrict the domain of $\alpha$ so that $\zeta^\star = -2 + \alpha b_{\hat{S}+1}$, and problem (18) reduces to

$$\begin{aligned}
\text{maximize} \quad & -\beta\alpha + (-2 + \alpha b_{\hat{S}+1}) + \sum_{s'=1}^{\hat{S}} \overline{p}_{sas'} - \sum_{s'=\hat{S}+1}^{S} \overline{p}_{sas'} \cdot \left( (-\alpha b_{s'} - 2 + \alpha b_{\hat{S}+1}) + \frac{(-\alpha b_{s'} - 2 + \alpha b_{\hat{S}+1})^2}{4} \right) \\
\text{subject to} \quad & \frac{2\sum_{s'=1}^{\hat{S}} \overline{p}_{sas'} + \left( \sum_{s'=\hat{S}+1}^{S} b_{s'} \overline{p}_{sas'} \right) \alpha}{\left( \sum_{s'=\hat{S}+1}^{S} \overline{p}_{sas'} \right)} \leq -2 + \alpha b_{\hat{S}+1} \\
& \alpha \in \mathbb{R}_+.
\end{aligned}$$

Note that the constraint in this problem is equivalent to

$$\frac{2\sum_{s'=1}^{\hat{S}}\overline{p}_{sas'} + \left(\sum_{s'=\hat{S}+1}^{S} b_{s'}\overline{p}_{sas'}\right)\alpha}{\left(\sum_{s'=\hat{S}+1}^{S}\overline{p}_{sas'}\right)} \leq -2 + \alpha b_{\hat{S}+1}$$

$$\Longleftrightarrow \quad 2\sum_{s'=1}^{\hat{S}}\overline{p}_{sas'} + \left(\sum_{s'=\hat{S}+1}^{S} b_{s'}\overline{p}_{sas'}\right)\alpha \leq \left(-2 + \alpha b_{\hat{S}+1}\right)\left(\sum_{s'=\hat{S}+1}^{S}\overline{p}_{sas'}\right)$$

$$\Longleftrightarrow \quad 2 \leq \left(\sum_{s'=\hat{S}+1}^{S}(b_{\hat{S}+1} - b_{s'})\overline{p}_{sas'}\right)\alpha$$

$$\Longleftrightarrow \quad 2\left(\sum_{s'=\hat{S}+1}^{S}(b_{\hat{S}+1} - b_{s'})\overline{p}_{sas'}\right)^{-1} \leq \alpha$$

Here, the last equivalence holds since the components of $b$ are sorted and $\sum_{s'=\hat{S}+1}^{S}(b_{\hat{S}+1} - b_{s'})\overline{p}_{sas'} \geq 0$. We have thus reduced the problem to a one-dimensional convex quadratic program with a box constraint, whose closed-form solution can be computed in $\mathcal{O}(1)$. Similar formulations can be derived for the other two cases of $\zeta^{\star}$.

In summary, the overall complexity is $\mathcal{O}(S\log S)$ due to the sorting of the components of $b$, while each of the $\mathcal{O}(S)$ subproblems can be solved in constant time $\mathcal{O}(1)$. $\qquad\square$

**Proof of Corollary 4.** The proof follows the same arguments as the proof of Corollary 3 and is therefore omitted. $\qquad\square$