# OpenReview forum: "Robust $\phi$-Divergence MDPs"
_NeurIPS.cc/2022/Conference — NeurIPS 2022 Accept_

### Official Review · Reviewer_gVaR · 2022-07-10

**Rating:** 5
**Confidence:** 4
**Soundness:** 3 good
**Presentation:** 3 good
**Contribution:** 2 fair

**Summary:**

The paper studies robust MDPs with s-rectangular ambiguity sets defined by $\phi$-divergence. The authors then propose a number of fast algorithms with complexity analyses for different popular $\phi$-divergences. Numerical results were then provided to support the methodologies.  The paper is well-written. The problem studied is highly relevant and of importance.

**Questions:**

- Why are $\phi$-divergence (s)-rectangular ambiguity sets useful?
- Can we handle ambiguity sets defined by several $\phi$-divergences?
- Can the algorithms developed apply to policy improvement?

**Ethics Review Area:**

["I don’t know"]

**Limitations:**

I don't see any limitations and potential negative societal impact of their work.

**Strengths And Weaknesses:**

### Strengths:

Robust MDP is an important topic in ML/AI and the question of how to quickly solve a robust MDP problem is highly relevant. It is known in the literature that to maintain tractability, one needs to stick with (s)- or (sa)-rectangular ambiguity sets. The paper focuses on several types of (sa)-rectangular sets and provides rigorous investigations, which is good and would be a good contribution to the research topic.

### Weaknesses:

My main concern goes to the novelty and the significance of the work. Robust MDP is already widely studied and well understood. The paper does not provide new robust MDP models, but only focuses on how to quickly solve the robust MDO problem under some well-known uncertainty settings (the use $\phi$-divergence ambiguity sets in robust MDP is not new). Note that $\phi$-divergence (sa)-rectangular ambiguity sets were already studied in the literature, thus the work here is just an extension to (s)-rectangularity settings.

**Other comments**

The paper only discusses value iteration; how's about policy evaluation and policy improvement?

The experiments are quite limited since the authors only compare their algorithms with MOSEK solvers. I can see that the Bellman equation has a standard max-min form with a linear objective function, for which several efficient algorithms exist.

Discussions on why $\phi$-divergence (s)-rectangular ambiguity sets are useful and how to construct them seem missing.

---

> ### Author Response · Authors · 2022-08-02
> **Response to the questions**
>
> 1. Why are $\phi$-divergence (s)-rectangular ambiguity sets useful?
>
> Please refer to our responses above.
>
> 2. Can we handle ambiguity sets defined by several $\phi$-divergences?
>
> This is an interesting suggestion! Yes, the proposed approach can handle the case of different $\phi$ functions in the summation of (2). If different $\phi$ functions are used in the ambiguity set (2), it would not change the overall outer bisection method, but different $\phi$ functions would lead to different projection problems, which we know how they could be solved. We now elaborate on this point in the paper; thank you!
>
> 3. Can the algorithms developed apply to policy improvement?
>
> Yes. By combining with techniques in [20], the developed algorithms can be generalized to solve policy improvement problems.
>
> *The numeric references above refer to the references in the initial submission of the paper.

---

> > ### Comment · Reviewer_gVaR · 2022-08-08
> > **Thank you for the responses**
> >
> > I thank the authors for the detailed responses that resolved many of my concerns/questions. Nevertheless, I am still not convinced by the use of MOSEK solvers. Since the Bellman equation forms a convex optimization problem, many gradient-based methods can be used (e.g. projected gradient descent) and can serve as baselines.

---

> > > ### Author Response · Authors · 2022-08-08
> > > **Thank you for reading our response!**
> > >
> > > Thank you very much for taking the time to read our response! We highly appreciate this!
> > >
> > > We would like to clarify that we compared to first-order method from [i].
> > >
> > > For solving problems (3) and (4) [main problems in our paper], we believe that gradient-based methods (or first order method) are not competitive to MOSEK solver or our proposed algorithms for the following reasons:
> > >
> > > 1. Unlike MOSEK solver or our proposed algorithms, first-order methods can not solve problems to high accuracy (efficiently). Since solving  (3) and (4) are only subroutines of the value iteration, we need to be able to solve to problems with high accuracy.
> > >
> > > 2. Using bisection method, the number of iterations is in O(log(length of the interval)), which is much less than the typical number of iterations used in first-order methods, even if the problem is not solved to high accuracy.
> > >
> > > 3. For the projection problem (4), the per-iteration complexities of our proposed algorithms are in the same order of the number of variables. For gradient-based methods, the complexity of evaluating the gradient at each iteration is at least in the same order of the number of variables.
> > >
> > > Therefore, we believe that first-order methods are not competitive in this particular setting. We are happy to explain this clearly in our camera ready, if the paper is accepted. We are also happy to compare with gradient-based methods (e.g. projected gradient descent) if you  have a strong interest in this direction.
> > >
> > > [i] Scalable First-Order Methods for Robust MDPs, 2021.

---

> ### Author Response · Authors · 2022-08-02
> **Response to the weaknesses**
>
> 1. Novelty and the significance of the work.
>
> Thank you for challenging us to clarify the contributions of our paper. To our best knowledge, even in the simpler setting of (s,a)-rectangularity there are no prior algorithms that solve RMDPs with general $\phi$-divergence ambiguity sets. Prior work solves special cases, such as total deviation ($L_1$), KL-divergence, or $L_2$-bounded ambiguity sets. Our new algorithms for dealing with $\phi$-divergence sets not only generalize this prior work into a single framework, but they also enable us to use new metrics, such as the Burg Entropy. Relying on the general framework of $\phi$-divergences, however, also introduces non-trivial algorithmic challenges, which we overcome through new theory and algorithms.
>
> We would also like to clarify why we focus on s-rectangular ambiguity sets. It has been shown that s-rectangular sets can provide policies that are less conservative (often by a factor of two or more) compared to (s,a)-rectangular sets [47], both in-sample and out-of-sample. This observation has motivated a recent surge in research on s-rectangular robust MDPs [5, 9, 15, 20]. We emphasize this fact in the revised paper (changes in blue). Unfortunately, solving the s-rectangular problem is more challenging because it involves searching over the continuous space of randomized policies, which implies that the robust Bellman operator no longer decomposes along actions.
>
> 2. How's about policy evaluation and policy improvement?
>
> Thank you! Our proposed framework speeds up the computation of the robust Bellman operator, which underlies most robust dynamic programming algorithms, including robust policy evaluation and robust policy improvement. Our accelerated Bellman operator can even be used in conjunction with value function approximation algorithms, such as robust projected value iteration and robust approximate policy iteration [i]. This is important as a naive computation of the robust Bellman operator as a convex program would have a practical complexity is $\mathcal{O} (S^3A^3)$, which would not scale to most problems of practical importance. We now clarify this point in the introduction.
>
> 3. The experiments are quite limited since the authors only compare their algorithms with MOSEK solvers.
>
> We apologize for the lack of clarity in the first draft of the paper. For the $\phi$-divergence ambiguity set, the associated optimization problem of the robust Bellman update contains nonlinear constraints. For the KL-divergence case, for example, the associated Bellman update is an exponential cone program. Therefore, linear optimization solvers are not suitable for our settings, and MOSEK appears to be the best available (commercial) solver for the problems of interest. We are also not aware of any tailored solution schemes for the generic robust Bellman update (3) that exploit the inherent structure. We now clarify this point in the introduction of the revised paper (changes in blue).
>
> 4. Discussions on why $\phi$-divergence (s)-rectangular ambiguity sets are useful and how to construct them seem missing.
>
> We apologize for this omission! The $\phi$-divergence ambiguity sets generalize a wide range of ambiguity sets that are considered in the related literature on distributionally robust and data-driven optimization, and they benefit from rigorous statistical performance guarantees. It has been shown recently, for example, that $\phi$-divergence constrained optimization problems are optimal among all (known and uknown) data-driven optimization paradigms if certain types of worst-case out-of-sample performance guarantees are being sought [ii]. The radii of $\phi$-divergence ambiguity sets can be selected either via cross-validation or via rigorous statistical bounds. We now discuss the advantages of $\phi$-divergence ambiguity sets in the introduction as well as the selection of their radii in the corresponding sections.
>
> As we discussed in our response to your first comment, we focus on s-rectangular ambiguity sets since it has been shown that they can provide policies that are less conservative (often by a factor of two or more) compared to (s,a)-rectangular sets [47], both in-sample and out-of-sample. This observation has motivated a recent surge in research on s-rectangular robust MDPs [5, 9, 15, 20]. We emphasize this fact in the revised paper. Unfortunately, solving the s-rectangular problem is more challenging because it involves searching over the continuous space of randomized policies, which implies that the robust Bellman operator no longer decomposes along actions.
>
> *The numeric references above refer to the references in the initial submission of the paper.
>
> [i] Tamar, Aviv, Shie Mannor, and Huan Xu. “Scaling up Robust MDPs Using Function Approximation.” In International Conference of Machine Learning (ICML), 2014.
>
> [ii] From Data to Decisions: Distributionally Robust Optimization Is Optimal, Management Science 2021.

---

### Official Review · Reviewer_KgfB · 2022-07-10

**Rating:** 5
**Confidence:** 2
**Soundness:** 3 good
**Presentation:** 3 good
**Contribution:** 3 good

**Summary:**

This paper proposed equivalence between the robust Bellman operator and a projected optimization problem. Also, several solutions to the projected optimization problems are developed for different distance measures.

**Questions:**

See the above part.

**Limitations:**

This paper mainly focuses on the model-based setting, which limits the contributions. Also, some external methods like binary search or other optimization methods are introduced and hence will complexify the method.

**Strengths And Weaknesses:**

Strengths: (1). The $\phi$-divergence is general hence several distance measures are included; (2). The proposed method works for many different uncertainty sets. (3). The complexity matches the one for non-robust problems.

Weakness: (1). Although this is a model-based method, many model-free methods aren't discussed, e.g., [1,2,3]. More related works should be discussed.
(2). Theoretically, the authors didn't explain clearly why the method in this paper is better than the value iteration or dynamic programming method for robust RL problems.



[1 ]Wang, Yue, and Shaofeng Zou. "Online robust reinforcement learning with model uncertainty." Advances in Neural Information Processing Systems 34 (2021): 7193-7206.
[2] Panaganti, Kishan, and Dileep Kalathil. "Sample Complexity of Robust Reinforcement Learning with a Generative Model." International Conference on Artificial Intelligence and Statistics. PMLR, 2022.
[3] Badrinath, Kishan Panaganti, and Dileep Kalathil. "Robust reinforcement learning using least-squares policy iteration with provable performance guarantees." International Conference on Machine Learning. PMLR, 2021.

---

> ### Author Response · Authors · 2022-08-02
> **Response to the limitations**
>
> 1. This paper mainly focuses on the model-based setting, which limits the contributions.
>
> Thank you. While this paper focuses on the model-based setting, we emphasize that efficient methods for computing robust Bellman updates extend to both approximate dynamic programming settings [44] and model-free settings [i, ii]. We focus on the model-based setting for two reasons. First, the model-based setting is often an important building block to constructing model-free algorithms in reinforcement learning. It appears reasonable to study the new setting of $\phi$-divergence based s-rectangular uncertainty sets first in the model-based context before translating them into the model-free context. While we expect that our algorithms can be extended to model-free settings, we leave this development to future work. Second, we would like to emphasize that the model-based setting itself also has many important real-life applications [iii-v], and that it is under active study in the machine learning community [vi-ix]. We now make this point explicit in the conclusions of the paper.
>
> 2. Some external methods like binary search or other optimization methods are introduced and hence will complexify the method.
>
> We note that all our algorithms are open-source and will be made available to the public (the URL is current withheld to maintain anonymity of the review process), as opposed to the commercial closed-source black-box optimizers (CPLEX, Gurobi, MOSEK) that are used otherwise. In some sense, we believe that this is comparable to e.g. the tailored solvers for support vector machines that are implemented in scikit-learn, which--despite having a non-trivial implementation--are not reliant on any third-party software and accessible as a mature technology to both researchers and practitioners.
>
> *The numeric references above refer to the references in the initial submission of the paper.
>
> [i] Reinforcement Learning under Model Mismatch, NIPS 2017.
>
> [ii] Robust Reinforcement Learning using Least Squares Policy Iteration with Provable Performance Guarantees, ICML 2021.
>
> [iii] Data uncertainty in Markov chains: Application to cost-effectiveness analyses of medical innovations. Operations Research, 2018.
>
> [iv] Distributionally robust inventory control when demand is a martingale. Mathematics of Operations Research, 2022.
>
> [v] Robust scheduling of EV charging load with uncertain wind power integration. IEEE Transactions on Smart Grid, 2018.
>
> [vi] Play to Grade: Testing Coding Games as Classifying Markov Decision Process, NeurIPS 2021.
>
> [vii] Fast Approximate Dynamic Programming for Infinite-Horizon Markov Decision Processes, NeurIPS 2021.
>
> [viii] Navigating to the Best Policy in Markov Decision Processes, NeurIPS 2021.
>
> [ix] MICo: Improved representations via sampling-based state similarity for Markov decision processes, NeurIPS 2021.

---

> ### Author Response · Authors · 2022-08-02
> **Response to the weaknesses**
>
> 1. Many model-free methods aren't discussed.
>
> Thank you for bringing the references [1, 2, 3] to our attention! Although we focus on the model-based setting, we agree that it is important to review at least some of the related model-free methods. The revised version of the paper discusses the suggested references in the introduction.
>
> 2. Why the method in this paper is better than the value iteration or dynamic programming.
>
> We would like to clarify that our proposed framework speeds up the computation of the robust Bellman operator, which underlies most robust dynamic programming algorithms, including robust value iteration and robust policy iteration. Our accelerated Bellman operator can even be used in conjunction with value function approximation algorithms, such as robust projected value iteration and robust approximate policy iteration [i]. This is important as a naive computation of the robust Bellman operator as a convex program would have a practical complexity is $\mathcal{O} (S^3A^3)$, which would not scale to most problems of practical importance. We now clarify this point in the introduction.
>
> [i] Tamar, Aviv, Shie Mannor, and Huan Xu. “Scaling up Robust MDPs Using Function Approximation.” In International Conference of Machine Learning (ICML), 2014.

---

### Official Review · Reviewer_19se · 2022-07-10

**Rating:** 6
**Confidence:** 5
**Soundness:** 2 fair
**Presentation:** 2 fair
**Contribution:** 2 fair

**Summary:**

This paper investigates the problem of computing the robust Bellman operator efficiently when the uncertainty set is defined using $\phi$-divergence. The basic idea is to calculate the robust Bellman update via simplex projections, which can be solved efficiently. Multiple examples of $\phi$-divergence defined uncertainty sets are given, including Kullback-Leibler Divergence, Burg Entropy, Variation Distance, $\chi^2$-divergence. Numerical experiments are provided to demonstrate the efficiency of the proposed method.

**Questions:**

Please see Strengths And Weaknesses.

**Limitations:**

NA.

**Strengths And Weaknesses:**

Strengths:
This paper investigates a more general notation of uncertainty set, s-rectangular, which is less conservative than the (s,a)-rectangular uncertainty set model. A large class of distance metrics, i.e., $\phi$-divergence defined uncertainty sets are studied. The proposed method is claimed to be more computationally efficient. Theoretical results and numerical experiements validates these claims. Overall, the contribution is useful in practice.


Weaknesses:
The major weakness of this paper lies in the presentation. In section 4, the author claim that the robust Bellman operator can be solved efficiently if the projection problem in (4) can be solved efficently. However, it is not discussed at all how the projection problem in (4) is related to the robust Bellman operator. It is thus not clear to this reviewer that once the projection in (4) is solved, how it is going to be used in the evaluation of the robust Bellman operator. As in section 5, the discussion is focused on solving the projection in (4) for various types of $\phi$-divergences. The overall presentation of the paper needs to be carefully revised.

Second, the setting focused in this paper is the s-rectangular uncertainty set. One question is that if the problem is reduced to the (s,a)-rectangular uncertainty set, is the challenging in evaluating the robust Bellman operator still exist? As for the (s,a)-rectangular uncertainty set, the problem is reduced to a distributionally robust optimization problem for each (s,a) pair. Therefore, the motivation of considering the s-rectangular uncertainty set shall be carefully discussed in order to motivate the research in this paper.

Third, this paper mainly focus on the model-based planning setting, where the uncertainty set is assumed to be known. Therefore, the approach in this paper might be difficult to be generalized to the practical reinforcement learning setting with large or even continuous state and action spaces.

Fourth, there is another line of research focus on model-free robust reinforcement learning, where the uncertainty set is not known, and samples from the center of the uncertainty set can be obtained. Different uncertainty set models are considered, e.g., ellipsoid type uncertainty set models and $\epsilon$-contamination uncertainty set model. Those algorithms are computationally efficient (the complexity at each iteration is independent of S and A), and can be easily extended to the case with function approximation for large state-action sapces.
Reinforcement learning under model mismatch. NIPS 2017.
Online Robust Reinforcement Learning with Model Uncertainty. NeurIPS 2021.
Robust reinforcement learning using least squares policy iteration with provable performance guarantees. ICML 2021.
Policy Gradient Method For Robust Reinforcement Learning, ICML 2022.

---

> ### Author Response · Authors · 2022-08-02
> **Response to the weakness (4)**
>
> 4. There are scalable algorithms for model-free robust reinforcement learning.
>
> Thank you for pointing out this line of research to us; although we focus on the model-based setting, we have updated the manuscript to include a brief survey of this literature stream. Please note that all of the aforementioned papers focus on (s,a)-rectangular uncertainty sets, whereas we study s-rectangular uncertainty sets. Also, to our best understanding, the complexity of the algorithms proposed by all four papers do depend on S and/or A, even though that complexity is somewhat less obvious:
>
> (a) In NIPS 2017, Section 4.1, the robust projected Bellman update requires the computation of the support function $\sigma$, which has a time complexity depending on S. In Section 4.2, to solve the mean squared projected Bellman equation, one needs to compute the gradient $\mu$, which has a time complexity (without applying any approximation) that depends on S, as stated in the paper.
>
> (b) In NeurIPS 2021, Algorithm 2, to update the parameters $\delta$ and $\theta$, the time complexity depends on S. For example, the summation needed in this step is over all states.
>
> (c) In ICML 2021, Algorithm 1 requires the computation of the equations (18)-(23). Equation (21) requires the computation of the support function $\sigma$, whose time complexity depends on S.
>
> (d) In ICML 2022, Algorithm 4, the computation of the B's and D's involves summations that depend on A. The computation of x requires the computation of an argmax over S components.

---

> ### Author Response · Authors · 2022-08-02
> **Response to the weakness (3)**
>
> 3. The approach might be difficult to be generalized to the practical reinforcement learning setting with large or even continuous state and action spaces.
>
> Thank you. While this paper focuses on the model-based setting, we emphasize that efficient methods for computing robust Bellman updates extend to both approximate dynamic programming settings [44] and model-free settings [i, ii]. We focus on the model-based setting for two reasons. First, the model-based setting is often an important building block to constructing model-free algorithms in reinforcement learning. It appears reasonable to study the new setting of $\phi$-divergence based s-rectangular uncertainty sets first in the model-based context before translating them into the model-free context. While we expect that our algorithms can be extended to model-free settings, we leave this development to future work. Second, we would like to emphasize that the model-based setting itself also has many important real-life applications [iii-v], and that it is under active study in the machine learning community [vi-ix]. We now make this point explicit in the conclusions of the paper.
>
> *The numeric references above refer to the references in the initial submission of the paper.
>
> [i] Reinforcement Learning under Model Mismatch, NIPS 2017.
>
> [ii] Robust Reinforcement Learning using Least Squares Policy Iteration with Provable Performance Guarantees, ICML 2021.
>
> [iii] Data uncertainty in Markov chains: Application to cost-effectiveness analyses of medical innovations. Operations Research, 2018.
>
> [iv] Distributionally robust inventory control when demand is a martingale. Mathematics of Operations Research, 2022.
>
> [v] Robust scheduling of EV charging load with uncertain wind power integration. IEEE Transactions on Smart Grid, 2018.
>
> [vi] Play to Grade: Testing Coding Games as Classifying Markov Decision Process, NeurIPS 2021.
>
> [vii] Fast Approximate Dynamic Programming for Infinite-Horizon Markov Decision Processes, NeurIPS 2021.
>
> [viii] Navigating to the Best Policy in Markov Decision Processes, NeurIPS 2021.
>
> [ix] MICo: Improved representations via sampling-based state similarity for Markov decision processes, NeurIPS 2021.

---

> ### Author Response · Authors · 2022-08-02
> **Response to the weaknesses (1) and (2)**
>
> 1. Presentation and the relationships between projection problem in (4) and the robust Bellman operator.
>
> We apologize for the lack of clarity in the initial draft of the manuscript. The equivalence between the robust Bellman update in (3) and the projection problem (4) is stated in Theorem 1, but in the previous version one needed to examine its proof to understand how the solution of (3) reduces to the repeated solution of (4). Based on your comment, we have revised our manuscript and uploaded a revision of our paper (changes in blue) that includes a dedicated description of this algorithm in the appendix, as well as a short discussion of this algorithm and signposting to the appendix in the main paper. Thank you for this suggestion!
>
> 2. How about (s,a)-rectangular uncertainty set and the motivation of considering the s-rectangular uncertainty set?
>
> Thank you; this is a great question! Solving (s,a)-rectangular RMDPs with $\phi$-divergence sets is also an unexplored and challenging problem, and our algorithms directly apply to this more specialized setting, too. We now point out in the revised paper that prior work on (s,a)-rectangular RMDPs with $\phi$-divergences have only addressed the subclass of KL-divergences [22, 29, 31]. We make this connection clearer in the revision. Our paper focuses on s-rectangular ambiguity sets for two reasons. First, it has been shown that s-rectangular sets can provide policies that are less conservative (often by a factor of two or more) compared to (s,a)-rectangular sets [47], both in-sample and out-of-sample. This observation has motivated a recent surge in research on s-rectangular robust MDPs [5, 9, 15, 20]. We emphasize this fact in the revised paper. Second, solving the s-rectangular problem is more challenging because it involves searching over the continuous space of randomized policies, which implies that the robust Bellman operator no longer decomposes along actions.
>
> *The numeric references above refer to the references in the initial submission of the paper.

---

> ### Comment · Reviewer_19se · 2022-08-08
> **after response summary**
>
> The authors have addresses most of my comments. I am not quite convinced by the discussion of weakness 3 i.e., extension to large/continuous state spaces. Therefore, I would keep my rateing of 6. weak accept.

---

### Author Response · Authors · 2022-08-09
**Thank you!**

We would like to thank all reviewers for their time reading this paper and providing reviews for our submission.

We take this last opportunity to clarify the importance of model-based methods. While we totally agree that model-free approach is an importance part of reinforcement learning, we would like to emphasize that model-based methods are complementary to model-free approach, and they have standalone merits.

In particular, many real-life settings are data-poor (few past observations relative to the size of the state/action spaces), and decisions are taken at a relatively slow time pace (e.g., inventory control where orders are taken weekly). In those problems, model-free methods will not work due to their data requirements. The situation resembles the use of linear regression (if few data points are available) vs. a neural network (when large amounts of data are available). One would not normally use a neural network in medical studies with a few hundred data points.

---

### Meta-Review · Area_Chair_SBDv · 2022-08-26

**Recommendation:** Accept
**Confidence:** Less certain

**Metareview:**

This is a somewhat borderline paper. The reviewers were unanimously positive, but they all had concerns. In reading through the concerns and responses, it seems that many (though perhaps not all) of the concerns could be addressed with additional references and some expository modifications.

If the paper makes the final cut, I encourage the authors to follow through with their proposed modifications and do their best to address the concerns expressed by the reviewers.

**Award:**

No

---

### Decision · Program_Chairs · 2022-09-14

Accept